# Upconversion amplification through dielectric superlensing modulation

Liangliang Liang [1], Daniel B.L. Teh [2,3], Ngoc-Duy Dinh [4], Weiqiang Chen[5], Qiushui Chen[1], Yiming Wu[1], Srikanta Chowdhury [6,7], Akihiro Yamanaka[6,7], Tze Chien Sum [5], Chia-Hung Chen[3,8], Nitish V. Thakor [3,4], Angelo H. All[9] & Xiaogang Liu [1,3,10]

Achieving efficient photon upconversion under low irradiance is not only a fundamental challenge but also central to numerous advanced applications spanning from photovoltaics to biophotonics. However, to date, almost all approaches for upconversion luminescence intensification require stringent controls over numerous factors such as composition and size of nanophosphors. Here, we report the utilization of dielectric microbeads to significantly enhance the photon upconversion processes in lanthanide-doped nanocrystals. By modulating the wavefront of both excitation and emission fields through dielectric superlensing effects, luminescence amplification up to 5 orders of magnitude can be achieved. This design delineates a general strategy to converge a low-power incident light beam into a photonic hotspot of high field intensity, while simultaneously enabling collimation of highly divergent emission for far-field accumulation. The dielectric superlensing-mediated strategy may provide a major step forward in facilitating photon upconversion processes toward practical applications in the fields of photobiology, energy conversion, and optogenetics.

[1] Department of Chemistry, National University of Singapore, Singapore 117543, Singapore. [2] Department of Biochemistry, National University of Singapore, Singapore 117456, Singapore. [3] Singapore Institute of Neurotechnology (SINAPSE), National University of Singapore, Singapore 117456, Singapore. [4] Department of Biomedical Engineering, National University of Singapore, Singapore 119228, Singapore. [5] Division of Physics and Applied Physics, School of Physical and Mathematical Sciences, Nanyang Technological University, Singapore 637371, Singapore. [6] Department of Neuroscience II, Research Institute of Environmental Medicine, Nagoya University, Nagoya 464-8601, Japan. [7] CREST, JST, Honcho Kawaguchi, Saitama 332-0012, Japan. [8] Department of Medicine, National University of Singapore, Singapore 117549, Singapore. [9] Department of Neurology, Johns Hopkins School of Medicine, Baltimore, MD 21205, USA. [10] Center for Functional Materials, National University of Singapore Suzhou Research Institute, Suzhou, Jiangsu 215123, China. Correspondence and requests for materials should be addressed to X.L. (email: chmlx@nus.edu.sg)

As a distinctive group of photon upconversion phosphors, lanthanide-doped upconversion nanoparticles (UCNPs) are capable of nonlinearly converting near-infrared (NIR) excitation into wavelength-programmable UV-Visible emission with large anti-Stokes shift, superior photostability, and long excited state lifetime[1,2]. However, given the inherently weak photon response of upconversion materials, a beam of laser light with coherent pump intensities above a certain threshold range is typically required to achieve detectable upconversion luminescence[3–7]. This is particularly true for UCNPs because their practical utilities in the biological and biomedical fields have been overly reliant on coherent excitation[8–11]. In recent years, a few implementations, based on surface passivation[12,13], surface plasmon coupling[14–17], photonic crystal engineering[18–20], and broadband sensitization[21–24], have led to experimental realizations with up to four orders of magnitude enhancement in upconversion emission intensity. Nevertheless, the conventional strategies have considerable limitations, such as the need for excitation at a specific wavelength, complex near-field configurations, critical atmosphere and humidity control, and stringent surface modification of the UCNPs. Direct realization of significant upconversion responses at subthreshold input intensities from an incoherent source remains an outstanding challenge.

Herein, we develop an approach that exploits bidirectional light confinement through the use of dielectric microbeads for giant upconversion luminescence enhancement. It should be noted that conventional dielectric optical nano-antennas (such as Si and Ge nanoparticles) feature high refractive indices and the ability to enhance the radiative emission rate of nearby emitters through Mie resonance[25–29]. By comparison, our approach is largely inspired by the fact that a transparent dielectric microbead can function as a superlens to confine an incident light beam into a sub-wavelength photonic hotspot with high local intensity for applications in super-resolution imaging[30–32], optical data storage[33,34], single nanoparticle detection[35,36], and nanojet lithography[37,38]. We reason that the dielectric microbead-guided wavefront modulation would be feasible for achieving efficient photon upconversion under low irradiance, as the high excitation fluence boosted by superlensing effects could surpass the pump threshold that is required for the practical realization of detectable upconversion.

## Results

**Spectroscopic study of upconversion amplification.** To validate our hypothesis, we fabricated polymeric microbeads made of highly transparent poly (ethylene glycol) diacrylate via a micro-fluidic technique and examined these microbeads for wavefront modulation (Fig. 1a). When two crossed LED light beams with green and orange color emission shined upon the microbeads obliquely, the beams passing through the microbeads converged into two small and bright spots on the substrate. (Fig. 1a inset). In a wide-field reflection mode, the subdiffractional groove features (~100 nm in width) of a blue-ray disk could be readily resolved through the combined use of the microbeads and a conventional optical microscope with a ×20 objective lens (Fig. 1b). These results suggest that the dielectric microbeads provide an excellent platform to enhance the local electric field intensity and confine the optical energy within the nanoscale domain[39].

To shed light on the wavefront-shaping feature of these polymeric microbeads, a three-dimensional finite-difference-time-domain (3D-FDTD) simulation was performed (Fig. 1c). For our simulation, we modeled a polymeric microbead (20 μm in diameter) monolayer resting on an UCNP-embedded polydimethylsiloxane (PDMS) substrate, and a periodic boundary condition was selected. Upon orthogonally polarized NIR

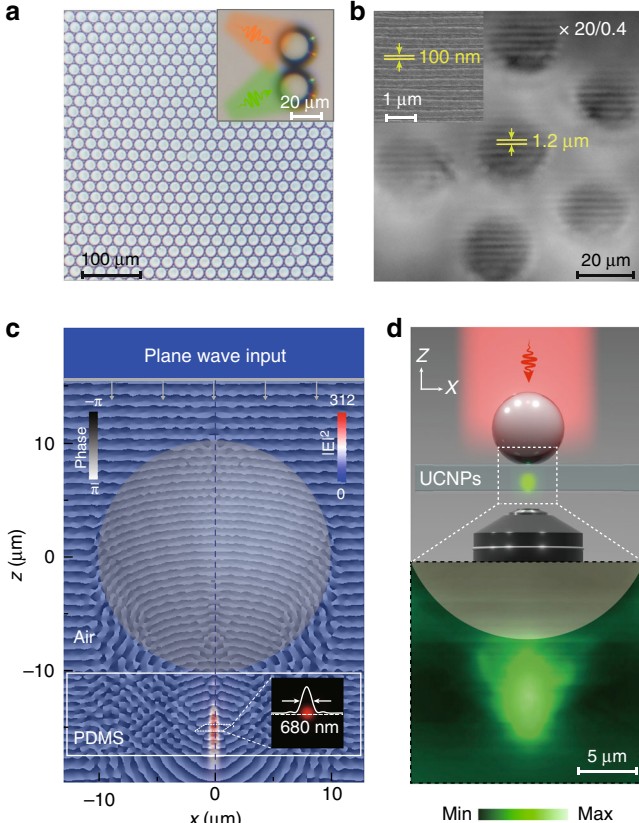

**Fig. 1** Superlensing effects of polymeric microbeads. **a** Photographic image of the as-prepared PEGDA polymeric microbeads using the microfluidic technique. (Inset) Photographic image showing the convergence of two intersected orange and green LED beams into two small focal spots after passing through the microbeads sitting on a glass substrate. **b** Microbead-guided high-resolution optical imaging of a blue-ray disk showing periodic line patterns of ~100 nm. (Inset) Scanning electron microscopic imaging of the disk. **c** FDTD simulation of the electric field distribution and phase variation of NIR excitation light after passing through the dielectric microbead. Note that a 980-nm plane wave light source is used for excitation and images of electric field distribution and absolute phase mapping are merged. (Inset) Cross-section feature of electric field at the focal spot. **d** Demonstration of the existence of the photonic hotspot in the UCNP-embedded polydimethylsiloxane (PDMS) film. Here, the NaYF$_4$:Yb/Er-embedded PDMS substrate is placed underneath a 20 μm dielectric microbead and a 980 nm laser beam is used for illumination. A confocal microscope was used to record the distribution of luminescence intensity through 3D scanning

plane-wave excitation at 980 nm, our simulation results showed that this polymeric microbead is able to confine the propagating incident energy from the far-field into a subwavelength photonic hotspot, with a full-width at half-maximum (FWHM) of about 680 nm, in the PDMS substrate. The intensity profiles derived from the cross-section analysis of the hotspot are highly symmetric, implying the reconstruction of a converging spherical wavefront to a focal point source, as verified by the simulated result of propagating phase mapping. The existence of the photonic hotspot was further experimentally confirmed by 3D confocal microscopic imaging of the spatial upconversion luminescence distribution of green-emitting UCNPs. Indeed, we recorded the formation of a green-emitting lobe, located near the rear side of the microbead, with dimensions spanning several micrometers in the PDMS substrate (Fig. 1d). Notably, the photonic crystal effect of the dielectric microbead monolayer was

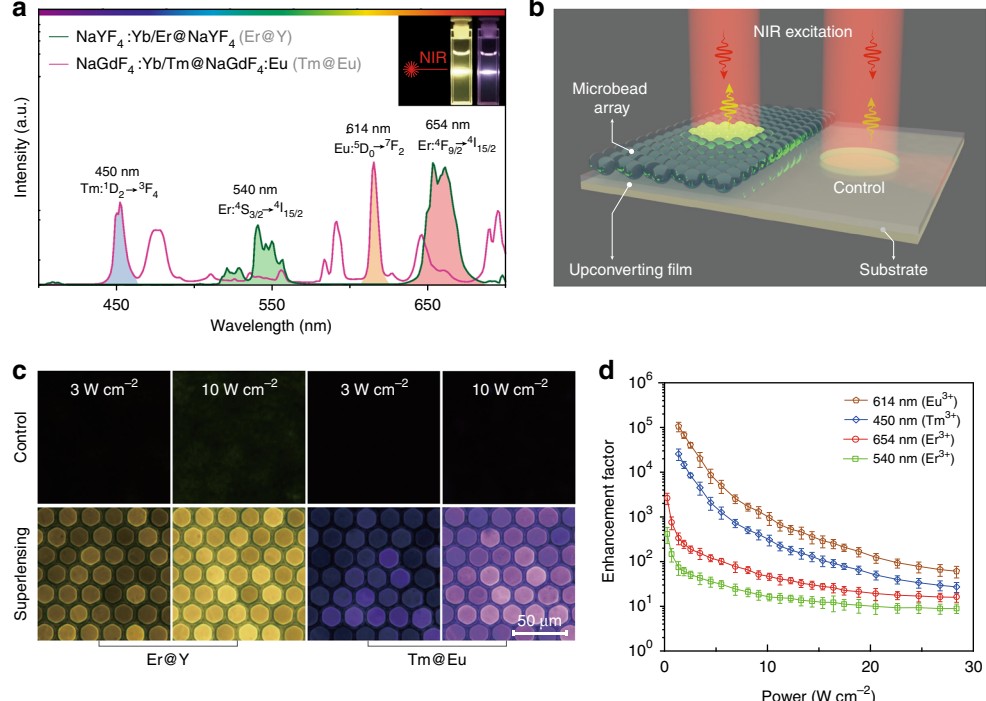

**Fig. 2** Experimental investigations on upconversion luminescence amplification through dielectric superlensing. **a** Upconversion emission spectra of NaYF$_4$:Yb/Er@NaYF$_4$ and NaGdF$_4$:Yb/Tm@NaGdF$_4$:Eu core-shell nanoparticles recorded under 980 nm laser excitation (~20 W cm$^{-2}$). (Inset) Corresponding photos of the nanoparticles dispersed in cyclohexane upon 980 nm excitation. **b** Schematic illustration of the experimental setup designed for luminescence amplification investigation. The as-prepared UCNPs are embedded in a PDMS precursor and then spin-coated on a glass substrate to form the upconverting film. The dielectric superlensing monolayer was prepared by adding polymeric microbeads on the top of the upconverting film. A ×10 objective lens was used to guide the 980 nm excitation light onto the film and simultaneously collect the upconversion emission. **c** Upconversion luminescence images of the UCNP-embedded PDMS upconverting films recorded with and without the microbead coverage upon 980 nm laser excitation at different intensities. **d** Power-dependent investigations of microbead monolayer induced-luminescence enhancement for PDMS films comprising NaYF$_4$:Yb/Er@NaYF$_4$ and NaGdF$_4$:Yb/Tm@NaGdF$_4$:Eu nanoparticles. Error bars represent ±1 s.d

also numerically investigated and no reflection or transmission band appear in the visible-to-NIR region (Supplementary Fig. 1).

In a further set of experiments, we synthesized NaYF$_4$:Yb/Er@NaYF$_4$ and NaGdF$_4$:Yb/Tm@NaGdF$_4$:Eu UCNPs of ~20 nm in diameter (Fig. 2a, Supplementary Fig. 2 and 3)[40,41]. Upon 980 nm excitation, Er$^{3+}$-activated UCNPs showed yellow-green emission with two characteristic bands centered around 540 and 654 nm. For UCNPs coactivated with Tm$^{3+}$/Eu$^{3+}$ dopants, characteristic bands centered around 450 and 614 nm were observed, arising from $^1D_2 \rightarrow {}^3F_4$ transition of Tm$^{3+}$ and $^5D_0 \rightarrow {}^7F_2$ transition of Eu$^{3+}$, respectively. It is important to note that the Eu$^{3+}$ emission stems from the energy-migration-mediated upconversion by accepting the down-shifting energy from Gd$^{3+}$ sublattice which is pre-populated by Tm$^{3+}$ activators in the core. To prove the effect of dielectric superlensing for enhanced upconversion processes, we prepared a monolayer array of hexagonal close-packed microbeads onto the UCNP-embedded PDMS substrate and compared the upconversion luminescence intensities of UCNPs with or without the microbead coverage (Fig. 2b, Supplementary Fig. 4). Figure 2c shows the upconversion luminescence images of the upconverting PDMS substrate containing UCNPs. The region covered with the microbeads showed bright visible emission, while only inconspicuous luminescence was captured in the absence of the microbeads (Supplementary Movie 1). Furthermore, we quantified the enhancement factor for the upconversion luminescence intensity of each activator under different power densities by comparing the peak intensities of its emission bands with and without the coverage of dielectric microbeads (Fig. 2d). Our data indicate that

the utilization of the dielectric microbeads can induce an immense luminescence enhancement in optical transitions of the activators under investigation. It is important to note that in all cases the luminescence-enhancing effect dominates to a much bigger extent under low-power excitation. We also observed that the luminescence enhancement features of dielectric superlensing are strongly dependent on the upconversion population processes of different activators. For instance, under 1.5 W cm$^{-2}$ power excitation, the upconversion luminescence at 614 nm could be enhanced by more than five orders of magnitude (up to 103,000) for the $^5D_0 \rightarrow {}^7F_2$ transition of Eu$^{3+}$. However, under the identical conditions, we recorded only ~100 fold-enhancement for emission band centered at 540 nm, corresponding to the $^4S_{3/2} \rightarrow {}^4I_{15/2}$ transition of Er$^{3+}$.

**Mechanistic investigation.** In a given lanthanide-mediated upconversion system, the sequential population of a specific emitting state is made possible through intermediate excited states[42,43]. In principle, the intensity of emission $I$, arising from an $n$-photon populated excited state, increases as a power function of excitation power $P$ ($I \propto P^n$) at the low-power end, and tends to reach a plateau at the high-power regime, because the excited state depopulation pathway is gradually dominated through population to higher excited states instead of relaxation to the ground state[43]. For illustration, let us consider the case of an UCNP-embedded PDMS substrate featuring dual activators (Tm@Eu). Power-dependent studies of luminescence enhancement revealed that the effect of pump fluence on emission intensity is more predominant under low power than that under

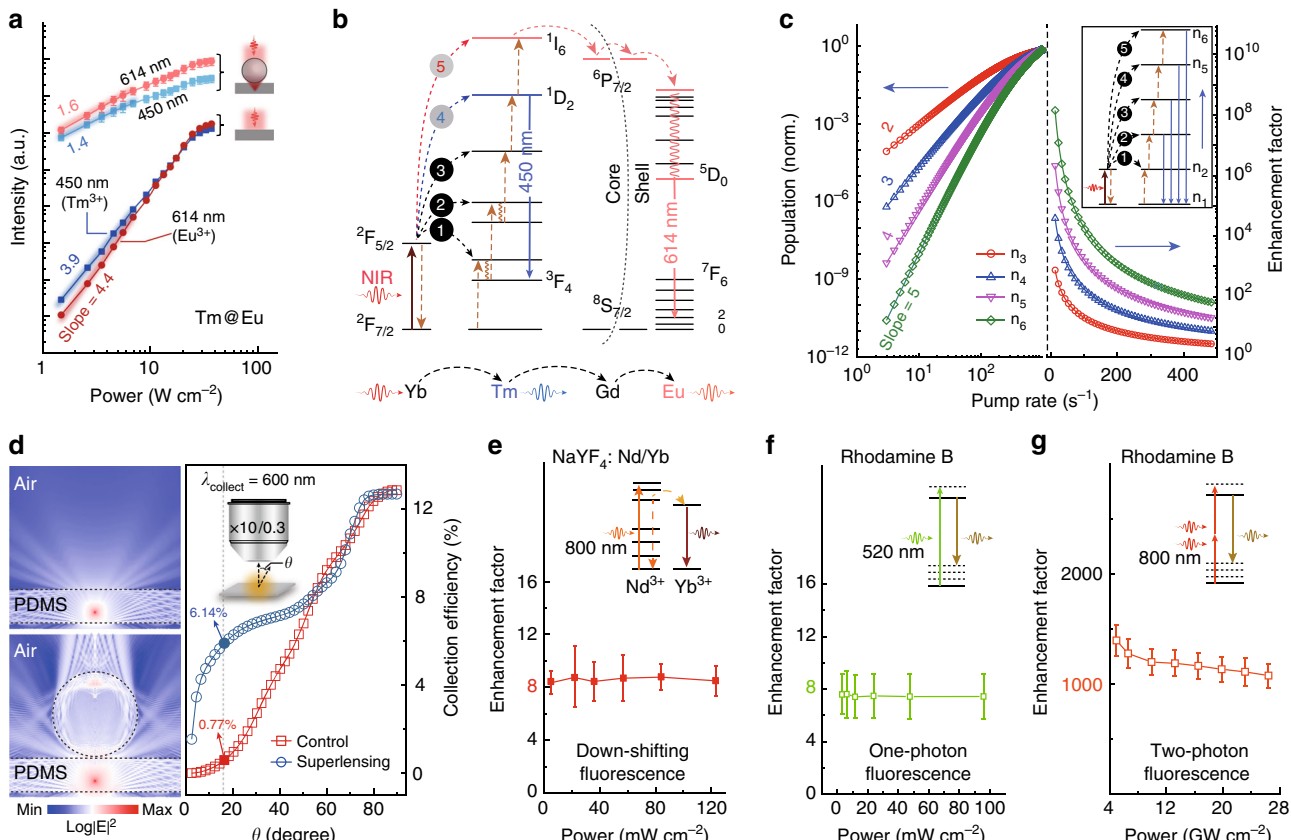

**Fig. 3** Mechanistic investigations of dielectric superlensing-mediated upconversion amplification. **a** Upconversion fluorescence power-dependent studies of luminescence enhancement in the $NaGdF_4:Yb/Tm@NaGdF_4:Eu$-embedded PDMS substrate, recorded with and without microbead coverage. Error bars represent ±1 s.d. **b** Proposed energy transfer diagrams showing the multistep excited state pumping of $NaGdF_4:Yb/Tm@NaGdF_4:Eu$ UCNPs. The emission of $Eu^{3+}$ is generated by accepting the down-shifting energy from $Gd^{3+}$ sublattice. **c** Numerical simulation results based on a simplified energy transfer model involving two-to-five-photon upconversion processes. It is clear to find that upconversion emission from higher excited states is more sensitive to the excitation pumping rate. Besides, the luminescence enhancement dominates with higher-order upconversion emission. **d** Comparative simulations of the far-field emission collection efficiency for upconversion enhancement, obtained in the presence or absence of a dielectric microbead. **e** Down-shifting luminescence enhancement of Nd/Yb codoped $NaYF_4$ nanocrystals using dielectric microbeads. (Inset) The proposed energy transfer mechanism for $NaYF_4:Nd/Yb$ down-shifting nanocrystals. Error bars represent ±1 s.d. **f**, **g** Microbead-mediated enhancement recorded for linear and two-photon absorption luminescence from Rhodamine B dye molecules. (Insets) The corresponding Stokes emission and two-photon absorption-emission mechanisms, respectively. Error bars represent ±1 s.d

high power excitation (Fig. 3a). At the lowest fluences, the emission intensities of $Tm^{3+}$ and $Eu^{3+}$ activators are largely increased. As the fluence increases, the enhancing factor for both activators becomes continuously smaller. However, when using dielectric microbeads to guide the excitation, the differences in the effect of power-dependent emission intensity could be brought down drastically, with much smaller increments in intensity at the low-power end than obtainable in the absence of the microbeads. This strongly suggests the presence of high excitation fluence occurring in UCNPs, which accounts for the main cause of enormous luminescence enhancement empowered by the microbeads.

The fluence dependence of a given upconversion transition is strongly dictated by its preceding population pathway[44,45]. For example, when comparing the $^1D_2 \rightarrow ^3F_4$ transition of $Tm^{3+}$ with the $^5D_0 \rightarrow ^7F_2$ transition of $Eu^{3+}$, one would recognize that $Eu^{3+}$ involves one additional step ($^1D_2 \rightarrow ^1I_6$) of population (Fig. 3b), shows higher excitation fluence sensitivity (Fig. 3a), and thus more significant luminescence enhancement induced by dielectric microbeads (Fig. 2d). Consistent with this, the green emission of $Er^{3+}$ at 540 nm, as already described in Fig. 2d, shows the lowest enhancement in upconversion luminescence, because its associated optical transition ($^4S_{3/2} \rightarrow ^4I_{15/2}$) involves only a two-step

population regime and thus has the lowest sensitivity to the excitation fluence. Numerically, we constructed a universal energy transfer model involving two to five-photon upconversion population (Fig. 3c, Supplementary Fig. 5). The simulation results reveal a clear trend in pump power sensitivity related to multiphoton upconversion process. Indeed, higher-order upconversion emission exhibits higher sensitivity to the excitation fluence and more prominent luminescence enhancement (Fig. 3c). At high excitation fluence, we observed a significant decrease in power sensitivity and a simultaneous decline in luminescence enhancement factor for all upconversion transitions under investigation.

Quite different from commonly used dielectric and metal optical nano-antennas for fluorescence enhancement by boosting radiative emission rate, our dielectric superlensing strategy for upconversion luminescence enhancement does not rely on the acceleration of radiative emission. According to the simulation results (Supplementary Fig. 6a, b), although a single dielectric microbead has a high-quality factor in whispering gallery mode, the large distance (>3 μm) between the dielectric microbead and emitting UCNPs around the focal point would make the potential Purcell effect negligible in affecting the spontaneous emission processes in UCNPs. For instance, although the upconversion

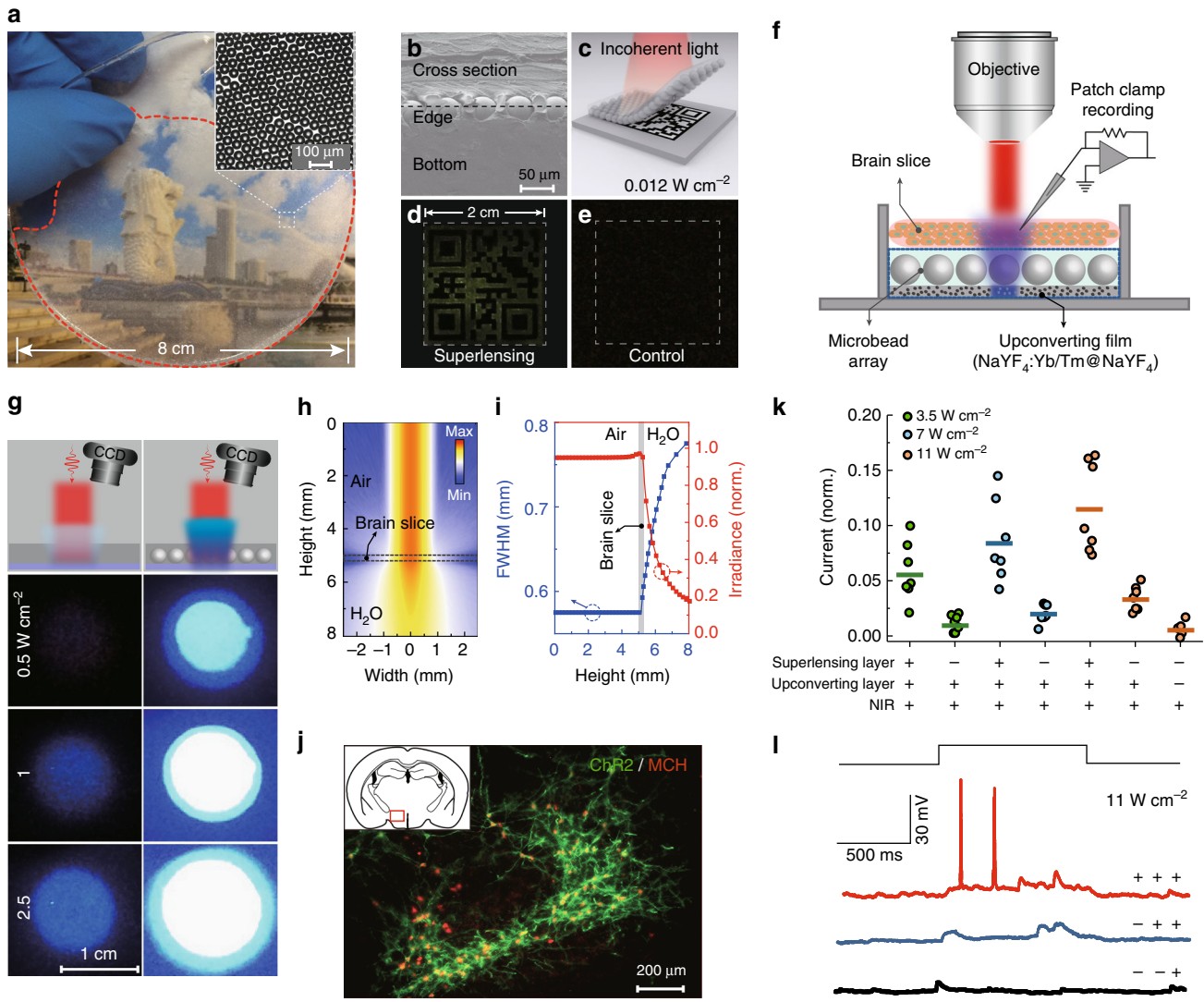

**Fig. 4** Scale-up of dielectric superlensing for advanced applications. **a** Photographic image of a PDMS composite sheet comprising 50 μm BaTiO₃ microbeads. Note that the distribution region of microbeads is circled with a red dashed curve. (Inset) Optical microscope image showing the ensemble of the microbeads used for making the planar sheet. **b–e** Demonstration of document security application using the composite sheet. The cross-section scanning electron microscopic image of composite sheet (**b**) and experimental design for upconversion-based encrypted barcoding (**c**). Photographic images, taken upon an incoherent light illumination, of an encrypted quick-response code with (**d**) and without (**e**) the composite sheet. **f** Schematic drawing illustrating the experimental setup for microbead-mediated optogenetics. Note that the setup consists of a microbead array and a thin layer of NaYF₄:Yb/Tm@NaYF₄ UCNPs. **g** Experimental setup and photographic images of the UCNP/PDMS layer (left panel) and the microbead/UCNP composite chip (right panel), recorded upon 980 nm excitation with varied power densities. **h, i** Monte Carlo simulation of NIR light (980 nm) attenuation and scattering of a brain slice (~300 μm thick). Cross-sectional view of the light intensity distribution of a collimated beam before and after penetration through the brain slice (**h**). Variation of the full width at half maximum (FWHM) and the irradiance of the light beam along the penetration path (**i**). **j** Fluorescence imaging of MCH neurons exclusively expressing ChR2 fused with eYFP (green) in the lateral hypothalamus. **k** Voltage-clamp tracing data showing the NIR/upconversion light-induced ion currents through the membranes of MCH neurons in an acute brain slice in response to 500-ms NIR stimulation at various intensities (*$p = 5.83 \times 10^{-4}$). Source data are provided as a Source Data file. **l** Comparison of action potential firing evoking capability of different PDMS chips on NIR illumination using current-clamp traces from MCH neurons in response to 1-s of NIR illumination at a power density of 11 W cm⁻². Here, these three +/− symbols represent the presence/absence of the dielectric superlensing layer, PDMS upconverting layer, and NIR irradiation

emission from Eu³⁺ (⁵D₀ → ⁷F₂) could be amplified up to five orders of magnitude, the dielectric superlensing effect introduced negligible changes in the fluorescence decay lifetimes throughout the whole emission band (Supplementary Fig. 6c, d). Therefore, the Purcell effect contributes negligibly to the overall upconversion luminescence enhancement under the current experimental design.

Apart from the ability to create photonic hotspots with high excitation field intensity, dielectric microbeads can enhance the efficiency of far-field light collection and in turn contribute to the emission amplification of photon upconversion to some extent. As shown in Fig. 3d, the wavefront of a point light source could be efficiently converted to a quasi-plane-wave and the divergent upconversion emission is well collimated into the far-field after passing through the dielectric microbeads (Supplementary Fig. 7). For example, in a divergence half-angle of about 17° with a numerical aperture of 0.3, our simulation shows that the use of a microbead leads to an approximately 8-fold increase in upconversion luminescence collection efficiency. Next, taking the NaYF₄:Nd/Yb nanocrystal emitting at ~980 nm from Yb³⁺

ions by accepting the energy from $Nd^{3+}$ ions as an example, we investigated dielectric superlensing effects for down-shifting fluorescence enhancement. The Stokes emission from $Yb^{3+}$ ions was amplified about eight times and the enhancement factors are independent of the excitation power intensity (Fig. 3e). The distinct fluorescence enhancement features for down-shifting fluorophores was further confirmed using Rhodamine B, which also showed a constant Stokes emission enhancement factor of 8 under a wide range of pumping power at 520 nm (Fig. 3f, Supplementary Fig. 8a). Therefore, although a large portion of one-photon emitters placed beneath the dielectric microbead is not illuminated, the overall enhancement in luminescence can still be observed benefiting from the superior light collection effect of the microbead. However, upon 800 nm femtosecond laser excitation, the anti-Stokes emission from Rhodamine B through the two-photon absorption could be amplified by a factor of more than 1100 (Fig. 3g, Supplementary Fig. 8b). Taken together, these data indicate a strong correlation between dielectric superlensing and nonlinear fluorescence amplification regardless of the type of fluorophores.

Our further experiments involving poly(ethylene glycol) diacrylate microbeads with a mean diameter of ~50 μm resulted in slightly inferior upconversion luminescence enhancement compared to that of 20 μm-sized microbeads (Supplementary Fig. 9). This might be due to incomplete polymerization for large-sized microbeads during UV-curing and the associated increase in focal length beyond the upconversion thin film. Despite the difference in physical parameters such as diameter, transmittance, and refractive index, commonly used dielectric microbeads made of polystyrene, poly(methyl methacrylate) or ceramic $SiO_2$ and $BaTiO_3$ materials could also be used to boost upconversion luminescence (Supplementary Fig. 10).

**Optogenetic application.** For convenience and improved mechanical strength, we combined an ensemble of close-packed, one-layer-thick $BaTiO_3$ microbeads (50 μm) with a PDMS precursor by curing, producing a transparent, planar composite PDMS sheet (Fig. 4a). This composite sheet showed superlensing effects similar to those of bare poly (ethylene glycol) diacrylate microbeads due to a good contrast in the refractive index between $BaTiO_3$ ($n = 2.1$) and PDMS ($n = 1.4$) (Supplementary Fig. 11). Importantly, the availability of this PDMS sheet allowed us to develop a convenient tool that can immediately apply to improve document security application involving upconversion-based encoding techniques. As shown in Fig. 4b–e, with the help of the as-fabricated PDMS sheet, an encrypted quick-response code, printed on a piece of paper using green-emitting NaYF₄:Yb/Er@NaYF₄ nanoparticle inks, could be readily decoded with a tungsten lamp at sub-solar irradiance (~0.012 W cm⁻²) (Supplementary Fig. 12). In contrast, in the absence of the PDMS sheet the decoding process was not successful. Interestingly, with this flexible dielectric superlensing film, imaging single UCNP or a PDMS film embedded with heavily $Tm^{3+}$-doped (4%) UCNPs could be realized under pumping power density of ~500 W cm⁻² and 80 W cm⁻², respectively, through the use of a common wide-field optical microscope (Supplementary Fig. 13)[46,47].

We next examined whether it would be possible to harness the effect of dielectric superlensing for advanced optogenetic applications[48,49]. The deep-tissue penetration ability of NIR light renders upconversion-mediated optogenetic stimulation a feasible solution for the remote, noninvasive optical interrogation of neurons with substantially improved spatial resolution and sensitivity. However, the recent implementation of upconversion-based optogenetics has been largely hindered by the requirement of high-power light excitation (higher than 200

W cm⁻²), as the strong intensity of light may cause harmful temperature rises in brain tissue[50,51]. A protocol involving low-power excitation would be extremely valuable for safe design of optogenetic experiments. Accordingly, we examined the combined utility of dielectric microbeads and NIR-mediated optogenetics for controlled neuronal depolarization and enhanced action potential firing. We first prepared a flexible PDMS chip that comprises layered $BaTiO_3$ microbeads and blue-emitting NaYF₄:Yb/Tm@NaYF₄ nanoparticles. We then placed the PDMS chip directly underneath a brain slice (~300 μm thick) that was connected to a patch clamp recording device. This design allows us to record the electrophysiological response of neurons containing melanin-concentrating hormone (MCH) (Fig. 4f, Supplementary Fig. 14). As shown in Fig. 4g, compared to the use of blue-emitting PDMS layer alone, the combined use of the UCNP/PDMS layer and the microbead monolayer led to a blue emission with significantly increased power output (Supplementary Movie 2). Although the placement of the 300 μm thick brain slice in the light path would cause the aberration of the irradiation beam from the laser to some extent (Fig. 4h, i), we confirmed the validity of our approach under low-power NIR excitation by the observation of neuronal activities with exclusive channelrhodopsin-2 (ChR2) expression (Fig. 4j). The photo-current ratios indicated that on 980 nm illumination at three different pump power settings, the dielectric superlensing effect boosted the current by 496% (3.5 W cm⁻²), 415% (7.0 W cm⁻²), and 340% (11 W cm⁻²), compared to controls without the microbead layer (Fig. 4k). These data imply that the use of dielectric microbeads may be able to overcome the current limitations associated with visible light scattering and attenuation in thick brain slices. At a low-power excitation of 11 W cm⁻², we were able to evoke a sufficient action potential to activate neuron cells, a feat which is inaccessible by conventional upconversion-based optogenetics (Fig. 4l).

## Discussion

In conclusion, our combined experimental and theoretical work has provided proof of the superlensing nature of a dielectric microbead suitable for boosting photon-photon interactions in UCNPs at sub-threshold light intensities. Unlike conventional strategies for upconversion amplification whose feasibilities are strongly dependent on many parameters of UCNPs such as particle size, composition, the choice of surface ligands, the approach described here provides a general solution for any given upconversion phosphors. Through this strategy, the magnitude of the upconversion response and, consequently, the physical process responsible for efficient upconversion can be controlled by modulating the wavefront of both excitation and emission fields through the use of dielectric microbeads. Our study also provides an experimental basis for facilitating two- or multiphoton pumped upconversion in organic systems. Importantly, the dielectric superlensing strategy may hold great promise in enabling triplet-triplet annihilation upconversion under sub-solar irradiance. In addition, our design is readily scalable for mass production and does not need stringent control over fabrication conditions or imaging atmosphere. These discoveries, when combined with recent advances in nanoscale synthesis and imaging technology, are expected to find applications in the fields of wide-bandgap solar cells, optical nanosensing, optical memory storage, biophotonics, and potentially many others.

## Methods

**Nanocrystal synthesis.** The designed NaYF₄:Yb/Er@NaYF₄ core-shell upconversion nanocrystals were synthesized following previously reported protocols with some modifications[40]. In a typical procedure, to a 50-mL two-neck round-bottomed flask charged with 1-octadecene (7 mL) and oleic acid (3 mL) was added

an aqueous solution (2 mL) of Y(CH$_3$CO$_2$)$_3$, Yb(CH$_3$CO$_2$)$_3$ and Er(CH$_3$CO$_2$)$_3$ at varied molar ratios (78/20/2) with a total lanthanide content of 0.4 mmol. After removal of the water by heating at 150 °C for 1 h, lanthanide complexes were formed. The mixture was cooled down to 50 °C, and a methanol solution (6 mL) of NH$_4$F (1.6 mmol) and NaOH (1 mmol) was subsequently added. After stirring for 30 min at 50 °C, the reaction was heated to 100 °C and pumped for 15 min to remove the methanol. The remaining residue was heated to 290 °C and then maintained at this temperature for 2 h under an argon flow. After cooling down to room temperature, the resulting nanocrystals were collected by centrifugation, washed with ethanol several times, and redispersed in 4 mL of cyclohexane for succeeding shell growth. To the synthesis of NaYF$_4$:Yb/Er@NaYF$_4$ core-shell nanocrystals, a shell precursor containing 0.4 mmol Y(CH$_3$CO$_2$)$_3$ was prepared by the same procedure as those mentioned above and cooled down to 80 °C. Then as-synthesized NaYF$_4$:Yb/Er core nanocrystals dispersed in cyclohexane was added. After 40 min, the resultant mixture was cooled down to 50 °C, and a methanol solution (6 mL) of NH$_4$F (1.6 mmol) and NaOH (1 mmol) was added subsequently. After stirring for another 30 min at 50 °C, the reaction was heated to 100 °C for 15 min before reaching 290 °C. After reaction completion in 2 h, the as-synthesized core-shell nanocrystals were isolated by centrifugation, washed with ethanol several times, and then redispersed in 4 mL cyclohexane.

The preparation of NaGdF$_4$:Yb/Tm@NaGdF$_4$:Eu core-shell nanocrystals follows the same procedure as mentioned above except for the use of different ratios of lanthanide ions. For the NaGdF$_4$:Yb/Tm core nanoparticles, the ratios of Gd/Yb/Tm was set at 50/49/1, and for the shell precursor, the ratio of Gd/Eu was set at 90/10.

**Preparation of polymeric microbeads**. For aqueous phase preparation, 1 mL water containing 0.4 mL PEGDA and 25 mg photo-initiator (2-hydroxy-4′-(2-hydroxythoxy)-2-methylpropiophenone) was sonicated for 5 min to yield a homogenous and transparent solution. For oil phase, 0.5% Krytox (modified) surfactant was added to fluorocarbon oil HFE 7500 to help to stabilize the microdroplets. The prepolymer microdroplets were polymerized by UV exposure (500 mW).

**Preparation of PDMS film with different phosphors**. Upconversion nanocrystals of 100 mg were redispersed in 2 mL CHCl$_3$ by sonication. Then 1.1 g PDMS (prepolymer and crosslinker in a 10:1 weight ratio) was mixed with the above solution and sonicated for 10 min to ensure a uniform distribution of UCNPs. Then the solution containing UCNPs was spin-coated onto a cover slide substrate at 7000 rpm for 1 min and the obtained film was solidified at 50 °C for 12 h. For the preparation of Rhodamine B imbedded PDMS film, Rhodamine B dye molecules (2 mg) was first dispersed in 2 mL CHCl$_3$, then 2.2 g PDMS (prepolymer and crosslinker in a 10:1 weight ratio) was mixed with the dye solution while stirring for 2 h. Then the mixture was spin-coated onto a cover slide at 10,000 rpm for 1 min and then kept at 50 °C in an oven for 12 h.

**Characterization**. The size and morphology of the as-prepared core-shell nanocrystals were characterized using a JEOL-1400 (Transmission electron microscope) TEM operating at an acceleration voltage of 100 kV. The powder (X-ray diffraction) XRD patterns were recorded by an ADDS X-ray diffractometer with CuKa radiation (40 kV, 40 mA, $\lambda = 1.54184$ Å). The luminescence spectra of samples dispersed in cyclohexane were recorded with an Edinburgh FSP920 spectrofluorometer equipped with a photomultiplier (PMT), in conjunction with a continuous-wave 980-nm diode laser (MDL-III). For film sample characterization, a home-built Olympus BX51 fluorescence microscope coupled with a 980-nm diode laser for excitation and an Ocean Optics Pro spectrometer (QE-Pro) for emission spectra collection is used. The divergence of the light beam was tuned with a convex lens system to make sure a collimated excitation laser beam can be obtained after passing through the objective lens (×10 , NA = 0.3). For measurements of simultaneous two-photon excited fluorescence of rhodamine B with and without the superlensing layer, a home-built fluorescence microscope coupled with a femtosecond amplified-pulsed laser system was used for excitation, and a visible monochromator (Acton, Spectra Pro 2750i) coupled with CCD (Princeton Instruments, Pixis 100B) was applied for emission collection. The excitation laser pulses were generated by a regenerative amplified femtosecond Ti:Sapphire laser system (~50 fs, 800 nm, 1 kHz; Libra, Coherent). The Coherent Libra regenerative amplifier was seeded by a femtosecond Ti:Sapphire oscillator (~50 fs, 80 MHz, Vitesse, Coherent). The temporal and spatial profiles of the applied excitation laser source follow a Gaussian distribution.

**Materials and methods for optogenetic study**. All experimental protocols involving animals were approved and performed in accordance by Institutional Animal Care and Use Committees, Research Institute of Environmental Medicine, Nagoya University, Japan.

**Acute brain slice preparation**. Male and female mice aged 3–4 months were used for electrophysiological recordings. Mice were deeply anesthetized using isoflurane (Wako Pure Chemical Industries, Osaka, Japan) and decapitated. The brain was then quickly isolated and chilled in ice-cold cutting solution (in mM: 110 K-gluconate, 0.05 EGTA, 15 KCl, 5 HEPES, 26.2 NaHCO$_3$, 3.3 MgCl$_2$, 25 glucose, and 0.0015 (±)-3-(2-carboxypiperazin-4-yl)propyl-1-phosphonic acid) gassed with 5%

CO$_2$ and 95% O$_2$. Coronal brain slices of 300 μm thickness that contain lateral hypothalamus were generated using a vibratome (VT-1200S; Leica, Wetzlar, Germany) and were temporarily placed in an incubation chamber containing bath solution (in mM: 3 KCl, 124 NaCl, 2 MgCl$_2$, 2 CaCl$_2$, 26 NaHCO$_3$, 1.23 NaH$_2$PO$_4$, and 25 glucose) gassed with 5% CO$_2$ and 95% O$_2$ in a 35 °C water bath for at least 60 min. Slices were then incubated in the same incubation chamber at room temperature for another 30–60 min for recovery. Cutting and bath solutions were modified from Pressler and Regehr.

**Electrophysiological recording using brain slice**. Acute brain slices of MCH-tTA; TetO ChR2 mice were transferred to a recording chamber (RC-26G; Warner Instruments, Hamden, CT, USA) on an upright fluorescence microscope (BX51WI; Olympus, Tokyo, Japan) stage and superfused with 95% O$_2$ and 5% CO$_2$ gassed bath solution at a rate of 1.2 mL/min using a peristaltic pump (Dynamax; Rainin, Oakland, CA, USA). An infrared camera (C3077-78; Hamamatsu Photonics, Hamamatsu, Japan) was installed in the fluorescence microscope along with an electron multiplying charge-coupled device (EMCCD) camera (Evolve 512 delta; Photometrics, Tucson, AZ, USA) and both images were separately displayed on monitors. Micropipettes of 4–7 MΩ resistance were prepared from borosilicate glass capillaries (GC150-10; Harvard Apparatus, Cambridge, MA, USA) using a horizontal puller (P-1000; Sutter Instrument, Novato, CA, USA). Patch pipettes were filled with KCl-based internal solution (in mM: 145 KCl, 1 MgCl$_2$, 10 HEPES, 1.1 EGTA, 2-Mg-ATP, 0.5 Na$_2$-GTP; pH 7.3 with KOH) with osmolality between 280 and 290 mOsm. Positive pressure was introduced in the patch pipette as it approached the cell. A giga-seal of resistance >1 GΩ was made between the patch pipette and the cell membrane by releasing the positive pressure upon contacting the cell. Then the patch membrane was ruptured by suction to form a whole-cell configuration. Electrophysiological properties of cells were monitored by an Axopatch 200B amplifier (Axon Instrument, Molecular Devices, Sunnyvale, CA). Output signals were low-pass filtered at 5 kHz and digitized at a sampling rate of 10 kHz. Patch clamp data were recorded through an analog-to-digital (AD) converter (Digidata 1550A; Molecular Devices) using pClamp software (Molecular Devices). Blue light of 475 ± 17.5 nm wavelength was generated by a light-emitting diode (Spectra light engine; Lumencor, Beaverton, OR, USA) and was guided to the microscope stage with a 1 cm diameter optical fiber. Near-infrared light of 976-nm wavelength was generated by LuOcean Mini 4 laser (Lumics GmbH, Berlin, Germany) and was guided by an optical fiber to illuminate brain slices.

Data are given as mean ± SEM. Nonparametric Mann–Whitney $U$-test was used for statistical analysis. All data were obtained from brain slices from 3 mices, where $n = 11$ cells for all beads experiment, $n = 8$ for all control experiments without beads, $n = 10$ for blue light LED experiments, and $n = 5$ for control experiments with only NIR excitation.

**Numerical simulation**. Simulations of the light converging effect of microbeads were performed by a three-dimensional finite-difference-time-domain (3D-FDTD) method with periodic boundary conditions. The light source for excitation was set as plane wave at a wavelength of 980 nm. For light harvesting ability simulation of upconversion emission, dipole source was used at a wavelength of 600 nm. The refractive index of PEGDA, PDMS, BaTiO$_3$ were set as 1.4, 1.4, and 2.1, respectively. For plane wave simulation, two simulations with orthogonally polarized beams were performed and the resulted fields were averaged incoherently. For dipole source simulation, three simulations with orthogonally polarized dipole sources were also performed. In order to evaluate the incoherent illumination situation, 11 models with light sources at different positions (varying over a region $(-\lambda/2, \lambda/2)$) were carried for simulation and the obtained light intensity distributions from each model were superimposed. To obtain the quality factor of dielectric microbead, a dipole source was placed on the surface of the microbead and an analysis group containing 36 time monitors was placed close to the surface of the microbead to record the decay of electric field. Additionally, Purcell factors of a dipole source were simulated when the dipole was placed inside the PDMS film with different distances (0.01–5 μm) from the microbead surface. During the simulation, a transmission box was applied to obtain the power emitted by the source.

The Monte Carlo method was used to simulate photon transport in a brain slice (300 μm in thickness) using the MATLAB software package mcxyz (http://omlc.org/software/mc/mcxyz/). The light source was modeled as a collimated beam with a waist of 0.5 mm. The source was placed on the top of the brain slice with 5 mm. This simulation domain was a cuboid with a length, width, and height of 0.5 cm, 0.5 cm and 0.8 cm, respectively. Tissue optical properties were set using the scattering and absorption parameters extracted from ref. [52], and the simulation time was set to 30 min.

For the numerical simulation of the sensitizer-activator upconversion system, we built a general sensitizer-activator upconversion system involving up to 5-photon upconversion population processes[53–55]. To show the universal features of the evolution of upconversion emissions, we kept the main upconversion transition processes (such as sensitizer absorption, energy transfer from sensitizer to activator, and intrinsic decay of excited states), and the secondary processes (such as activator absorption, energy back transfer from activator to sensitizer, and cross-relaxation between activators) were not taken into consideration. In addition, the energy gap between adjacent main excited states was set as equal, and the

upconversion coefficients between sensitizers and each active state of activators shared the same value. Thus, the following rate equations for a typically simplified sensitizer-activator upconversion system was established.

$$n_{s2}^{'}[t] = P_{980}n_{s1}[t] - w_s n_{s2}[t] - \\ (c_1 n_1[t] + c_2 n_2[t] + c_3 n_3[t] + c_4 n_4[t] + c_5 n_5[t])n_{s2}[t] \tag{1}$$

$$n_{s1}^{'}[t] = -n_{s2}^{'}[t] \tag{2}$$

$$n1'[t] = -c_1 n_{s2}[t]n_1[t] + w_2 n_2[t] + b_{31}w_3 n_3[t] \\ + b_{41}w_4 n_4[t] + b_{51}w_5 n_5[t] + b_{61}w_6 n_6[t] \tag{3}$$

$$n2'[t] = c_1 n_{s2}[t]n_1[t] + b_{32}w_3 n_3[t] + b_{42}w_4 n_4[t] \\ + b_{52}w_5 n_5[t] + b_{62}w_6 n_6[t] - w_2 n_2[t] - c_2 n_{s2}[t]n_2[t] \tag{4}$$

$$n3'[t] = c_2 n_{s2}[t]n_2[t] + b_{43}w_4 n_4[t] + b_{53}w_5 n_5[t] \\ + b_{63}w_6 n_6[t] - w_3 n_3[t] - c_3 n_{s2}[t]n_3[t] \tag{5}$$

$$n4'[t] = c_3 n_{s2}[t]n_3[t] + b_{54}w_5 n_5[t] + b_{64}w_6 n_6[t] \\ - w_4 n_4[t] - c_4 n_{s2}[t]n_4[t] \tag{6}$$

$$n5'[t] = c_4 n_{s2}[t]n_4[t] + b_{65}w_6 n_6[t] - w_5 n_5[t] \\ - c_5 n_{s2}[t]n_5[t] \tag{7}$$

$$n_6^{'}[t] = c_5 n_{s2}[t]n_5[t] - w_6 n_6[t] \tag{8}$$

where $P_{980}$ is the absorption rate of sensitizer; $w_i$ is the intrinsic excited state decay rate of activators; $w_s$ is the intrinsic decay rate of sensitizers; $b_{ij}$ is the branching ratio for activator decaying from excited state $i$ to $j$; $c_i$ is the upconversion coefficient between the sensitizer and activator residing at the excited state $i$. The steady state population density evolution of each excited state ($n_3$, $n_4$, $n_5$, $n_6$) of activators was obtained by solving the rate equation at different $P_{980}$ values. It should be noted that the modeling conducted here means to build a universal sensitizer-activator upconversion model involving essential transition processes for general lanthanide-doped upconversion, so that the simulation results can reveal the general trends of population evolution regardless of the selection of sensitizers or activators.

**Reporting summary.** Further information on experimental design is available in the Nature Research Reporting Summary linked to this article.

## Data availability
All relevant data that support the findings of this work are available from the corresponding author upon reasonable request. The source data underlying Fig. 4k and Supplementary Fig. 14 d are provided as a Source Data file.

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

## Acknowledgements

This work is supported by the Singapore Ministry of Education (Grant R143000627112, R143000642112), Agency for Science, Technology and Research (A*STAR) (Grant R143000684305), National Research Foundation, Prime Minister's Office, Singapore under its Competitive Research Program (CRP Award No. NRF-CRP15-2015-03), National Basic Research Program of China (973 Program, Grant 2015CB932200), Japan Science and Technology Agency CREST (Grant JPMJCR1656), the CAS/SAFEA Inter-national Partnership Program for Creative Research Teams, and National Natural Science Foundation of China (21771135, 21471109). T.C.S. acknowledges the support from JSPS-NTU Joint Research Project M4082176, the Ministry of Education AcRF Tier 1 grant (RG173/16) and Tier 2 grants (MOE2015-T2-2-015 and MOE2016-T2-1-034), the Singapore National Research Foundation through the Competitive Research Programme (NRF-CRP14-2014-03), and the NRF Investigatorship Programme (NRF-NRFI-2018-04).

## Author contributions

L.L. and X.L. conceived and initiated the project. X.L., A.H.A., N.V.T., C.H.C., T.C.S., and A.Y. supervised the project and led the collaboration efforts. L.L and Y.W synthesized the upconversion nanocrystals and performed theoretical simulations. L.L. and W.C. conducted the optical experiments. N.D.D., Q.C., and L.L. prepared polymeric microbeads. D.B.L.T. and S.C. conducted the optogenetic experiments. L.L. and X.L. wrote the manuscript. All authors participated in the discussion and analysis of the manuscript.
