## [Peer Review File · Nature Communications]

Reviewers' comments:

Reviewer #1 (Remarks to the Author):

Photon upconversion process in rare earth doped luminescent nanoparticles have attracted a broad range of interests from multidisciplinary research communities. The goal is to pursue high efficiency and high brightness, at low irradiance of excitation. Towards this goal, in the year of 2018 alone, there are already several major developments published by Nature x journals, including the revisit of dye-sensitization concept by utilizing the triplet state to facilitate upconversion enhancement (ref 18), two independent works around building core-shell-shell design of single UCNPs to allow highly doped sensitizers and activators (<https://www.nature.com/articles/s41566-018-0217-1>; <https://www.nature.com/articles/s41467-018-05577-8>), and the activation of the surface dark-layer to enhance upconversion in a thermal field (<https://www.nature.com/articles/s41566-018-0108-5>). This paper suggests a new practical strategy for upconversion enhancement. I quite enjoy reading this manuscript, because of its new knowledge discovered in using superlens to significantly and non-linearly improve the higher photon excitation states of the multi photon upconversion emissions, and its practical utility of self-assembled superlens (microsphere) arrays in a large scale flexible film. In fact, this inspiring work should be able to attract interests from multidisciplinary research communities, e.g. photonics, nanotechnology, and material sciences. I suggest the recent developments of upconversion enhancements should be captured in the introduction section of this manuscript. Together with this work, the field of upconversion is booming!

I suggest the following changes to improve the quality of this work.

1. My main suggestion is to promote the data and presentation of Extended Data Figure 6, as they are the 'eye-catching' part that photonics community is less familiar with, otherwise the current data and discussions around the main text Figure 1 are fairly straightforward, less important.
2. Superlens, because of the size of each microspheres, there are dead areas where the UCNPs can not be illuminated. There are only a few of UCNPs being illuminated after each microsphere. So the actually enhancement factors could be a lot of larger than the reported data for same amount of luminescent UCNPs. Whether it is possible to detect single UCNP, with or without a microsphere?
3. This also means microspheres used in this work (20 μm and 50 μm) at different size will significantly affect the number of UCNPs being illuminated. The authors reported that "Our control experiments involving poly (ethylene glycol) diacrylate microbeads with a mean diameter of 50 μm resulted in luminescence enhancement comparable to that of 20 μm -sized microbeads (Extended Data Fig. 7)." Is this true? The inconsistency of the microspheres at different sizes and index needs attention in terms of their optics path, working distance and focal areas, then in principle, should be different.
4. In Extended Data Figure 8, why shows the core@shell-like luminescent microparticles in PS and SiO₂? Is it because the selective absorption of the microparticles or the varied excitation power at different position?
5. There are different units used for the power density, such as W/cm², mW/mm², and mJ/cm². It is better to use same unit to make the results comparable.
6. I would be curious to know if the superlens strategy could significantly enhance the brightness of highly doped UCNPs, e.g. 4 %- 8% Tm doped UCNPs, if that works, it certainly provides a new strategy for single NP imaging (see two of our recent developments, "Microscopic inspection and tracking of single upconversion nanoparticles in living cells", 7, e18007; doi: 10.1038/lisa.2018.7, and "Multi-photon near-infrared emission saturation nanoscopy using upconversion nanoparticles", DOI : 10.1038/s41467-018-05842-w.". They are highly dependent on the excitation irradiance, and in principle, the superlens strategy could solve a key bottleneck issue for the single NP imaging)
7. As for a fixed objective lens, the collection efficiency should be much higher when the divergence half-angle is smaller than the aperture angle. Why it was different in Fig. 3d?
8. As shown in Fig. 3f and Extended Data Fig. 6c, normally, the two-photon absorption process is nonlinear in the whole region, which may cause large error to obtain Fig. 3f using the extrapolated

data. For example, the enhancement factor of two-photon upconversion process (540 nm Er³⁺) is only 20~100, while the two-photon absorption process is 1100. Besides, the error bar in Extended Data Fig. 6 is quite large. Some discussions are needed here.

9. More descriptions and details are needed for the figure captions of Fig. 4g and 4h. For example, what are these curves in Fig. 4h represent for, especially the order of + and - ?

Overall, a great concept, and experimental investigations, with potential broad impacts to the science and technology community! Appreciate this level of contributions to the field!

Reviewer #2 (Remarks to the Author):

In this article, the authors are using dielectric polymer materials to enhance upconversion emission and demonstrating the usage of the enhanced fluorescence in optogenic applications. However, the idea that using dielectric material to enhance fluorescence can derive back to 2000s. Besides, the written is not easy to follow. Besides, the written is not easy to follow. Some points are even over-concluded. The organization of the figure is not systematic, e.g. figures in different group show the same idea.

1. The idea of this article is not novel enough to meet criteria of Nat Comm. There were several reports on dielectric materials to enhance fluorescence recent years, such as Nano Lett., 2016, 16 (8), pp 5143–5151 <https://pubs.acs.org/doi/abs/10.1021/acs.nanolett.6b02076> and Phys. Chem. Chem. Phys., 2016, 18, 19324-19335 <https://pubs.rsc.org/en/content/articlehtml/2016/cp/c6cp03303b>. None of these articles have been cited.
2. The article is lack of deep discussion on the mechanism of enhancement. In page 4, it says 'We also observed that the enhancement effects of dielectric superlensing are strongly dependent on the specific nature of upconversion transitions.' As the enhancement of fluorescence have been studied before with other luminescent materials (see papers in point 1), the enhancement may not related to upconversion nature at all.
3. The figure captions are not well written. The arrangement of figure is very messy and not self-consistent.
4. The authors firstly use PEG diacrylate for experiment to show dielectric materials for wavefront modulation, while later use PDMS for simulation. As the dielectric constant for PEG and PDMS is largely different, the materials used for experimental demonstration should be same as the simulation materials.
5. Since the authors used several types of dielectric materials for demonstration, the effect of permittivity should be discussed to be more comprehensive. Transmission, reflection and refractive index of each material should be provided.
6. What is quality factor of the polymeric microbeads array in Fig 1a? Is the figure taken in bright field by microscope?
7. As the total upconversion emission is enhanced, how is the life time affected by the modification?
8. In line 1, paragraph 2 of Page 2, the terminology 'bidirectional wavefront modulation' is not common in the field of optics and photonics.
9. In Page 4, 'We also observed that the enhancement effects of dielectric superlensing are strongly dependent on the specific nature of upconversion transitions.' This is not clear to me.
10. In page 4, 'we quantified the enhancement factor for the characteristic emission peaks of each activator (Fig. 2d).' there maybe spectrum overlap/interference between different doping ions, how to separate them and eliminate the interference before quantification? How is Fig2d obtained?
11. The method to quantify the enhancement factor for each activator is not clearly stated.

12. The authors should provide short explanation on simulation method in the main text rather than just state in the supplementary files. Why the method is used?
13. Periodic boundary condition should be used, instead of 'FDTD simulation', to simulate the enhancement effect. Photonic crystal effect should be considered.
14. On page 6, it says 'Our control experiments involving poly (ethylene glycol) diacrylate microbeads with a mean diameter of 50 μm resulted in luminescence enhancement comparable to that of 20 μm -sized microbeads (Extended Data Fig. 7).' The authors should discuss about what is the point of the statement.
15. The caption Fig 3c is not clear.
16. The reference should be added after the exact item rather than aggregate together after one sentence if several reference are cited.

Reviewer #3 (Remarks to the Author):

In this work, the authors propose an elegant method for enhancing the light intensity emitted by upconversion nanoparticles (UCNPs). Upconversion is a multiphoton process that has low probability and therefore that needs in principle high excitation power density. This aspect is hampering the use of the otherwise highly performant UCNPs in a wealth of applications including high-resolution bioimaging. Until now, several strategies have been tested to increase the luminosity of UCNPs, including plasmonics, broadband excitation, and optical crystals. In related fields, some researchers have developed local mini-lenses to concentrate the excitation beam (e.g. solar pumped YAG lasers, see Motohiro et al., Jap. J. Appl. Phys. 2017, DOI) or when it comes to microscopy, sper lenses in order to overcome the diffraction limit and to reach resolution on the order of 100 nm (see Bing Yan et al., Appl. Optics 2017, 46, 3142). Here the authors transpose these ideas in the microworld by developing polymeric microbeads generating a superlensing effect. Irradiation of the beads with a NIR light results in a concentrated light beam (< 1 micrometer in diameter) that efficiently excites underlying UCNPs. In turn, the upconverted light can be used as a light source for exciting downshifted emission of other lanthanide ions, lending to the system a welcome versatility. The design is clever and the authors convincingly demonstrate that it indeed works and that the intensity enhancement obtained for the upconversion light reaches five orders of magnitude. This is one order of magnitude larger than previously reported devices. Overall the microbead/UCNPs proposed systems are adequately characterized and their properties investigated with respect to several parameters. Convincing examples are also given. As conclusion I think that this work is worth publication in Nature Communications after the following minor points are taken into consideration:

1. In the discussion section, it would be nice could the authors compare their proposed system with the enhancement systems published in the literature, not only with respect to performances but, also, with respect to ease of design and practical use.
2. Page 2, Last section, 2nd sentence. Please rephrase this sentence; firstly, green and orange light are not intrinsically "invisible"; what the authors mean is that they use a low power so that these beams are not detectable by naked eye. Secondly, I do not understand how these two beams "self-assemble"; this is not a correct way of describing the interaction between two light beams. Indeed self-assembly means that two or more components interact to give a more complex architecture. It is not the case here; self-assembly cannot qualify the interaction of two light beams – if they interact at all, because the generated bright spots (visible to the naked eye?) seem to have the same color than the initial beams.

3. Figure 2a, ED Figure 2 and corresponding text:

I have some difficulty with qualifying the luminescence of europium as being "upconverted light". The original definition of upconversion calls for a process in which an atom (ion) absorbs two (or more) low-energy photons and generates an anti-Stokes emission of one higher energy photon. This is obviously not the case for Tm@Eu in which the upconversion occurs for Tm(III) which then

transfers the excitation energy to Gd(III) and Eu(III). To avoid some confusion, the authors should clearly state that they use an extension of the upconversion concept, in that the emitting ion is not the one absorbing two (or more) photons. Or more correctly that Eu(III) emission is activated by the up-converted light of Tm(III).

4. Figures 3a,b. Mechanism of energy transfer:

According to Figure 3a, the slope of the Tm-upconverted emission versus excitation power is about 4 as is the slope of the Eu(III) emission; therefore why in Figure 3b the mechanistic path is going through the Tm(¹I₆) level that necessitates the absorption of 5 photons? Can one also consider a radiative excitation from the Tm(¹D₂) level?

What happens if Gd(III) is replaced with non-luminescent Lu(III)?

What is the explanation for the slope diminishing considerably (to around 1.5) when microbead coverage is introduced?

5. ED Figure 6a and associated text page 5.

"Down-conversion" should not be used here in that it is defined as the "inverse" of upconversion (absorption of a high-energy photon leading to emission of two or more lower-energy photons).

"Down-shifting" would be the appropriate term.

6. Surface power unit

Four different units are used throughout the manuscript which may be confusing, e.g. W/cm²; mW/cm²; W/mm²; mW/mm²; at least the latter two should be replaced with the first two. In particular, p. 7, while discussing optogenetic applications, an excitation power of 110 mW/mm² is mentioned; this translates into 11 W/cm², a value far above the accepted safe range for biological tissues.

Review Matrix

<Responses to the reviewers' comments (Manuscript number NCOMMS-18-23530-T)

Title: Upconversion Amplification through Dielectric Superlensing Modulation

Reviewer(s)' Comments to Author:

Referee: 3

Comments to the Author

Changes in the revised manuscript as a response to the reviewers' comments are highlighted in red color and clarifications regarding the reviewer's comments are provided in blue color.

Reviewer #1:

Photon upconversion process in rare earth doped luminescent nanoparticles have attracted a broad range of interests from multidisciplinary research communities. The goal is to pursue high efficiency and high brightness, at low irradiance of excitation. Towards this goal, in the year of 2018 alone, there are already several major developments published by Nature x journals, including the revisit of dye-sensitization concept by utilizing the triplet state to facilitate upconversion enhancement (ref 18), two independent works around building core-shell shell design of single UCNPs to allow highly doped sensitizers and activators (<https://www.nature.com/articles/s41566-018-0217-1>; <https://www.nature.com/articles/s41467-018-05577-8>), and the activation of the surface dark-layer to enhance upconversion in a thermal field (<https://www.nature.com/articles/s41566-018-0108-5>). This paper suggests a new practical strategy for upconversion enhancement. I quite enjoy reading this manuscript, because of its new knowledge discovered in using superlens to significantly and non-linearly improve the higher photon excitation states of the multi photon upconversion emissions, and its practical utility of self-assembled superlens (microsphere) arrays in a large scale flexible film. In fact, this inspiring work should be able to attract interests from multidisciplinary research communities, e.g. photonics, nanotechnology, and material sciences. I suggest the recent developments of upconversion enhancements should be captured in the introduction section of this manuscript. Together with this work, the field of upconversion is booming!

Response: We sincerely thank the reviewer for the careful evaluation and high opinion of our work. Indeed, we feel that the development of practical approaches for significant upconversion fluorescence enhancement is important and could further stimulate the practical application of lanthanide-doped UCNPs in many emerging areas such as super-resolution imaging, ultrasensitive bio-detection, and lasing. More importantly, the dielectric superlensing strategy for upconversion fluorescence enhancement proposed here could even be extended to fluorescence enhancement on other type of nonlinear processes such as triplet-triplet annihilation fluorescence and two-multiphoton absorption fluorescence.

I suggest the following changes to improve the quality of this work.

Comments #1: My main suggestion is to promote the data and presentation of Extended Data Figure 6, as they

are the ‘eye-catching’ part that photonics community is less familiar with, otherwise the current data and discussions around the main text Figure 1 are fairly straightforward, less important.

Response: We thank this reviewer for the suggestion. In the revised manuscript, to make the it more reader-accessible and easy to understand, we have reorganized the figures in Extended Data Figure 6 (Supplementary Figure 8 in the revised version) and Figure 3. In addition, we also modified the related presentation and figure captions in the revised manuscript. We believe this modification makes the discussion more straightforward.

Comments #2: Superlens, because of the size of each microspheres, there are dead areas where the UCNP's can not be illuminated. There are only a few of UCNP's being illuminated after each microsphere. So the actually enhancement factors could be a lot of larger than the reported data for same amount of luminescent UCNP's. Whether it is possible to detect single UCNP, with or without a microsphere?

Response: We thank this reviewer for this valuable remark. According to the suggestion, we have conducted a number of experiments and demonstrated that highly dispersed UCNP's could be easily detected using our superlensing strategy (Figure RL 1, also included in Supplementary Figure 13). As shown in Figure RL 1.a, NaYF₄:Yb/Er@NaYF₄ (20/2%) core-shell UCNP's were firstly prepared and spin-coated on the surface of a silicon wafer. Due to slight particle aggregation, nano-aggregates such as UCNP-tetramers instead of monodispersed UCNP's were observed, we call these nano-aggregates quasi-single UCNP. Then a conventional wide-field microscope coupled with a color CCD camera and a grating spectrometer was used to image these quasi-single UCNP's. Upon an ultralow irradiance of 500 W cm⁻² at 980 nm, upconversion emission spectrum and spatial dispersion of these quasi-single UCNP's could be clearly detected (50 ms integrating time) with the help of the dielectric superlensing layer (Figure RL 1, c and d), however, no signal could be obtained when the dielectric-superlensing layer is not used (Figure RL 1, b and d). These additional data could further evidence the impact of our strategy for practical applications such as single-particle imaging, single molecule detection and potentially many others.

Figure RL 1. Demonstration of quasi-single UCNP imaging with dielectric-superlensing monolayer. **a**, SEM image of prepared NaYF₄:Yb/Er@NaYF₄ (20/2%) core-shell UCNP's spin-coated on a silicon wafer. **b**, Wide-field microscopic imaging of quasi-single UCNP's without using and **c**, using the BaTiO₃/PDMS composite film. **d**, The obtained upconversion emission spectra from **b** and **c**.

Comments #3: This also means microspheres used in this work (20 μm and 50 μm) at different size will significantly affect the number of UCNPs being illuminated. The authors reported that “Our control experiments involving poly (ethylene glycol) diacrylate microbeads with a mean diameter of 50 μm resulted in luminescence enhancement comparable to that of 20 μm -sized microbeads (Extended Data Fig. 7).” Is this true? The inconsistency of the microspheres at different sizes and index needs attention in terms of their optics path, working distance and focal areas, then in principle, should be different.

Response: We thank the reviewer for pointing out this issue, largely due to our unclear writing. In fact, our experiments involving poly(ethylene glycol) diacrylate microbeads with a mean diameter of $\sim 50 \mu\text{m}$ resulted in slightly better upconversion luminescence enhancement compared to that of 20 μm -sized microbeads (see Supplementary Figure 9). We totally concur with the possible reasons pointed out by the reviewer. In addition, this might be due to several other possible factors such as larger microdroplet dimension-induced incomplete polymerization during UV-curing, larger dead area for UCNPs in the PDMS film, and longer focal length which might extend out of the upconversion film sample. Following the reviewer’s suggestion, a discussion has been provided in the revised manuscript.

Comments #4: In Extended Data Figure 8, why shows the core@shell-like luminescent microparticles in PS and SiO_2 ? Is it because the selective absorption of the microparticles or the varied excitation power at different position?

Response: The core@shell-like luminescent microparticles in PS and SiO_2 is interesting. After comparison, we can find that this phenomenon only occurs on Tm@Eu samples those emit blue and pink color. The reason pointed out by the reviewer is possible. In addition, as small PS and SiO_2 microbeads may have lower composition homogeneity throughout the whole microsphere, refractive indexes on their outer-shell close to the surface might be different with those of their inner part. This may lead to the spatial separation of blue and pink emission after passing through the microbeads, resulting in the core@shell like luminescent microparticles.

Comments #5: There are different units used for the power density, such as W/cm^2 , mW/mm^2 , and mJ/cm^2 . It is better to use same unit to make the results comparable.

Response: We thank the reviewer for the notice. We have modified the power density units to W cm^{-2} accordingly in the revised manuscript.

Comments #6: I would be curious to know if the superlens strategy could significantly enhance the brightness of highly doped UCNPs, e.g. 4 %- 8% Tm doped UCNPs, if that works, it certainly provides a new strategy for single NP imaging (see two of our recent developments, “Microscopic inspection and tracking of single upconversion nanoparticles in living cells”, 7, e18007; doi: 10.1038/lisa.2018.7, and “Multi-photon near-infrared emission saturation nanoscopy using upconversion nanoparticles”, DOI : 10.1038/s41467-018-05842-w.”. They are highly dependent on the excitation irradiance, and in principle, the superlens strategy could solve a key bottleneck issue for the single NP imaging)

Response: We totally agree with the reviewer on the fact that an approach could significantly enhance the brightness of highly doped UCNPs could solve a key bottleneck issue for single nanoparticle imaging. Following the reviewer’s comment, with $\text{NaYF}_4:\text{Yb}/\text{Tm}$ (20/4%) core UCNPs as an example, we have demonstrated the significant upconversion luminescence enhancement on highly doped UCNPs (Figure RL 2, also included in Supplementary Figure 13). As shown in Figure RL 2 a, upon 980 nm laser excitation, no upconversion signal could

be detected from the PDMS film containing these highly doped UCNPs. However, when the dielectric superlensing film was introduced, using a wide-field optical microscope, the strong blue upconversion fluorescence could be easily captured (50 ms of integration) upon an extremely low NIR irradiance of 80 W cm^{-2} . This additional experiment further evidenced the superior performance of our dielectric superlensing film in upconversion fluorescence amplification. More importantly, combining with bright UCNPs, the dielectric superlensing approach may make single-nanoparticle imaging a big step forward.

Figure RL 2. Demonstration of the upconversion fluorescence enhancement for highly doped UCNPs using dielectric superlensing. **a**, Wide-field microscopic imaging of highly doped UCNPs without using and **b**, using the BaTiO₃/PDMS composite film. Inset in **b** is the collected upconversion emission spectra from **a** and **b**.

Comments #7: As for a fixed objective lens, the collection efficiency should be much higher when the divergence half-angle is smaller than the aperture angle. Why it was different in Fig. 3d?

Response: In our experimental setup, an objective lens with an aperture half angle of $\sim 17^\circ$ was used. According to our simulation results in Figure 3d, after passing through the surface of the PDMS film, only less than 1% of upconversion light could be collected in this divergence half-angle. However, when a $20 \mu\text{m}$ microbead was placed on the top of the film, large portion of upconversion light (more than 6%) could be effectively guided and collimated into the above-mentioned aperture half angle in the far-field, resulting in enhanced collection efficiency of upconversion fluorescence.

Comments #8: As shown in Fig. 3f and Extended Data Fig. 6c, normally, the two-photon absorption process is nonlinear in the whole region, which may cause large error to obtain Fig. 3f using the extrapolated data. For example, the enhancement factor of two-photon upconversion process (540 nm Er³⁺) is only 20~100, while the two-photon absorption process is 1100. Besides, the error bar in Extended Data Fig. 6 is quite large. Some discussions are needed here.

Response: We thank the reviewer for raising this concern. Actually, although upon high power excitation, the slope extracted from the power-dependence of the two-photon absorption fluorescence of Rhodamine B is still 1.99 (see Supplementary Figure 8 b in the revised manuscript), which is extremely close to the theoretical value of 2. However, for nonlinear processes of lanthanide-doped UCNPs, the slope would decrease very quickly along with the increasing of excitation power. Therefore, using the extrapolation strategy to obtain the two-photon absorption fluorescence intensity of Rhodamine B in the low power excitation region may not necessarily result in a large error of measurement. As for the luminescence enhancement factor of two-photon upconversion process of Er³⁺, this value can also achieve ~ 1000 folds when the excitation intensity is low. However, due to the gradually

decreased value of slope, the enhancement factor would be reduced to less than 100 folds at a higher excitation power density.

It should be noted that, the distribution of nanoparticles and especially hydrophilic dye molecules in PDMS film is not absolutely homogeneous, which may lead to the difference in the number of emitters exposed inside photonic hotspots and the error during the testing. In addition, the error would appear more obvious when the linear axis is used.

Comments #9: More descriptions and details are needed for the figure captions of Fig. 4g and 4h. For example, what are these curves in Fig. 4h represent for, especially the order of + and -?

Response: Following the reviewer's suggestion, we have added more details in the figure captions of Figure 4g and h in the revised manuscript.

Comments #10: Overall, a great concept, and experimental investigations, with potential broad impacts to the science and technology community! Appreciate this level of contributions to the field!

Response: We greatly appreciate the positive comments and efforts made by this reviewer on our work.

Reviewer #2:

In this article, the authors are using dielectric polymer materials to enhance upconversion emission and demonstrating the usage of the enhanced fluorescence in optogenetic applications. However, the idea that using dielectric material to enhance fluorescence can derive back to 2000s. Besides, the written is not easy to follow. Besides, the written is not easy to follow. Some points are even over-concluded. The organization of the figure is not systematic, e.g. figures in different group show the same idea.

Response: We sincerely thank the reviewer's efforts and critical comments on our manuscript. However, with all due respect, we feel that the reviewer has underestimated the novelty and importance of our work, perhaps because of our unclear presentation in our first submission.

We agree with the reviewer on the fact that dielectric materials have been used for fluorescence enhancement. Comparing with plasmonic metal (such as Ag and Au) nanostructures, dielectric nanomaterials (~ 100 nm) featuring high refractive indexes ($n > 3$) such as silicon and germanium nanoparticles could also act as optical antennas supporting spectra resonances for fluorescence enhancement application. However, the dielectric superlensing approach for photon upconversion fluorescence enhancement we demonstrated in our manuscript is quite different from the traditional enhancement approaches boasted by dielectric optical nano-antennas. Specifically, we list below some key points to show the differences:

- In our work, we use polymer material with very low refractive index of ~ 1.4 to fabricate the micron-sized dielectric superlenses with diameter of about 20 μm . Therefore, due to the low refractive index and ultra-large dimension, these dielectric superlenses would not support spectra resonances like traditional dielectric optical nano-antennas in the visible region. In our additional experiments, we didn't observe noticeable reduce in lifetime for upconversion fluorescence from Eu^{3+} although whose fluorescence was amplified up to five orders of magnitude. Thus, quite different from the situations involving dielectric nano-antennas for fluorescence enhancement, our dielectric superlensing mediated photon upconversion amplification is not due to the increased emission rate of emitters induced by the spectra overlap between upconversion emission band and the resonance band.
- Different from dye molecule-based Stokes emission, lanthanide-doped upconversion is a unique nonlinear optical process in which the fluorescence intensity is exponentially related to the excitation power, especially in the low power region. Although dielectric nano-antennas could also contribute some degrees of linear enhancement in Stokes fluorescence when their resonance band overlaps with the excitation region of phosphors, the nonlinear nature of upconversion can make the fluorescence of UCNPs exponentially sensitive to the variation in local excitation power density (*Phys. Rev. B.* 2000, 61,3337; *Phys. Rev. B.* 2005, 71, 125123). Furthermore, without the need for stringent nanofabrication techniques (such as electron beam lithography and reactive ion etching) to create dielectric nano-antennas with a specific resonance band in the excitation region, the dielectric superlenses could easily confine the incident excitation energy into a subwavelength photonic hotspot with significantly enhanced field density. As shown in our manuscript, with the help of the dielectric superlensing film, the fluorescence enhancement of stokes emission is less than 10. However this value can even achieve more than 100,000 for upconversion process. Therefore, the dielectric superlensing strategy is quite practical and has a strong correlation with photon upconversion fluorescence enhancement.
- Moreover, our dielectric superlensing strategy for fluorescence enhancement demonstrated here is not limited to lanthanide-doped UCNPs, this approach could be easily extended to other nonlinear optical processes such as triplet-triplet annihilation and multiphoton absorption fluorescence.

In closing, the dielectric superlensing strategy for upconversion enhancement presented in our work is quite different with the traditional dielectric nano-antennas in the aspects of material properties, enhancement mechanism, practicality, performance, and targeted luminescent materials. We hope the reviewer concurs after going through our clarification.

Comments #1: The idea of this article is not novel enough to meet criteria of Nat Comm. There were several reports on dielectric materials to enhance fluorescence recent years, such as Nano Lett., 2016, 16 (8), pp 5143–5151 <https://pubs.acs.org/doi/abs/10.1021/acs.nanolett.6b02076> and Phys. Chem. Chem. Phys., 2016, 18, 19324–19335 <https://pubs.rsc.org/en/content/articlehtml/2016/cp/c6cp03303b>. None of these articles have been cited.

Response: As suggested, we have added a couple of closely related references in our revised manuscript.

Comments #2: The article is lack of deep discussion on the mechanism of enhancement. In page 4, it says ‘We also observed that the enhancement effects of dielectric superlensing are strongly dependent on the specific nature of upconversion transitions.’ As the enhancement of fluorescence have been studied before with other luminescent materials (see papers in point 1), the enhancement may not relate to upconversion nature at all.

Response: We understand the reviewer’s concern on the mechanism of upconversion fluorescence enhancement, largely due to the lack of a detailed introduction to the unique properties of upconversion luminescence in the introduction section. In addition, we also thank the reviewer for the valuable suggestions of taking additional experiments to investigate the possible effect from photonic crystal and change of radiative decay rate.

Lanthanide-doped upconversion nanocrystals enable anti-Stokes emission upon near-infrared excitation. For an upconversion process to proceed, lanthanide ions such as Er^{3+} , Tm^{3+} , and Ho^{3+} are typically selected as activators and Yb^{3+} ion with larger absorption cross-section is usually co-doped as sensitizer. When illuminated with near-infrared light, incident low-energy photons can primarily be absorbed by Yb^{3+} sensitizer and then the energy can be transferred to adjacent activators that are homogeneously confined in the crystal lattice for sequential population at higher excited states and finally high-energy photons would be released (*Nat. Nanotechnol.* 2015,10, 924; *Chem. Rev.* 2004, 104, 139). Therefore, for a typical upconversion emission of one high energy photon, two or more low energy pumping photons are absorbed. Due to this intrinsic nonlinear nature of upconversion fluorescence, the fluorescence intensity is exponentially instead of linearly sensitive to the excitation power density. Meanwhile, the fluorescence from upconversion process involving more low-energy photons could be much more sensitivity to excitation power density.

Thus, benefiting from the nonlinear nature, increasing the excitation rate should also be a very effective way for upconversion fluorescence enhancement. Firstly, we demonstrated the strong excitation light confinement capability of prepared dielectric superlenses both theoretically and experimentally to evidence that these superlenses could help to create photonic hotspots with extremely high local excitation power density (see Figure 1). Next, we experimentally showed that upconversion fluorescence could be efficiently enhanced and enhancement factor could be quite different for different upconversion processes (see Figure 2). This phenomenon was further well explained using numerical simulation based on a simplified energy transfer upconversion model (Figure 3c). All these results indicate that the dielectric superlensing induced fluorescence enhancement properties is strongly related to the upconversion processes.

To further support this, we studied the fluorescence enhancement based on NaYF₄:Nd/Yb and even Rhodamine B dye (Figure 3e and f), quite different from the upconversion fluorescence, the down-shifting fluorescence could only be enhanced by 8 times (due to light collection effect induce by superlens) regardless of the excitation power density. However, the two-photon absorption fluorescence of Rhodamine B could be enhanced more than 1100 folds. These results strongly evidenced the strong correlation between dielectric superlensing strategy and photon upconversion processes.

Figure RL 3. Upconversion fluorescence decay lifetime investigation on NaGdF₄:Yb/Tm@NaGdF₄:Eu UCNPs with and without the dielectric superlensing film.

More importantly, in the revised manuscript, to study whether the significant upconversion emission enhancement is induced by the increased radiative rate of emitters, we investigated the fluorescence lifetime variation of Eu³⁺ (614 nm) with and without adding the dielectric superlenses. As can be seen from Figure RL 3 (also see Supplementary Figure s6), the fluorescence decay lifetime of Eu³⁺ show negligible change although its upconversion fluorescence showed highest enhancement factor of more than five orders of magnitude. Thus, we can conclude that the significant upconversion fluorescence enhancement induced by dielectric superlensing is not due to the enhanced radiative rate of emitters.

Additionally, following the reviewer’s suggestion, we also considered the photonic crystal property of these dielectric superlens array, our simulation results showed no reflection/transmission band could be observed in the visible range (see Figure RL 4). Thus, the influence of photonic crystal effect of dielectric superlens array could also be excluded.

Based on all these explorations above, we believe the significant upconversion fluorescence enhancement induced by dielectric superlensing is mainly due to the intrinsic nonlinear optical property of UCNPs and strong excitation light confining capability for creating photonic hotspots featuring high local power density. We hope these additional experiments could further help to validate the mechanism of dielectric superlensing-induced photon upconversion enhancement.

Figure RL 4. Photonic crystal property study of a close-packed dielectric superlens array. The transmission spectrum is obtained using FDTD simulation.

Comments #3: The figure captions are not well written. The arrangement of figure is very messy and not self-consistent.

Response: To make the figure captions more readable to readers from different disciplines, more details have been added in figure captions in the revised manuscript. Figure 2d and Figure 3 have also been rearranged for clarity.

Comments #4: The authors firstly use PEG diacrylate for experiment to show dielectric materials for wavefront modulation, while later use PDMS for simulation. As the dielectric constant for PEG and PDMS is largely different, the materials used for experimental demonstration should be same as the simulation materials.

Response: I believe the reviewer made a mistake here. For our simulation, PEGDA superlens was placed on the top of a PDMS film containing UNCPS. Thus, for the simulation of light confining effect of excitation light, a planewave at 980 nm was used to simulate the incident excitation light from the top of the dielectric superlens. Inversely, a dipole source (600 nm) placed in the PDMS film was used to simulate the upconversion emitter.

Comments #5: Since the authors used several types of dielectric materials for demonstration, the effect of permittivity should be discussed to be more comprehensive. Transmission, reflection and refractive index of each material should be provided.

Response: We agree with the reviewer that the difference in permittivity, transmission, and reflection of microbeads could result in different fluorescence enhancement. In our work, due to the high stability and ease of fabrication, PEGDA microbeads were used to demonstrate this efficient approach for upconversion fluorescence enhancement. However, the selection of specific material for practical application should be determined by many factors such as surface condition, stability, size, and surrounding medium. More discussion has been added in the revised manuscript, and refractive indices of each material were added in the figure caption.

Comments #6: What is quality factor of the polymeric microbeads array in Fig 1a? Is the figure taken in bright field by microscope?

Response: The figure was taken in bright field by a conventional optical microscope. During picture-taking, the as-prepared microbeads were immersed in the oil phase and self-assembled in the microfluidic channel. Meanwhile, a white light source from the top was used for illumination. To take the inserted picture, microbeads were firstly placed on the top of a glass substrate (surrounding is air), then two crossed LED light beam with green and orange color emission were used to shine the dielectric microbeads with a tilted angle about 60° to the substrate. We can clearly find that both the light beams were well confined to form two small and bright spots on the substrate. In our revised manuscript, related details have been added and the figure in Figure 1a has been redesigned to make it clearer.

Additionally, we also did FDTD simulation to study the quality factor of single PEGDA polymeric microbeads (20 μm in diameter). According to the simulation (Figure RL 5), a resonance peak around 980 nm (978.9 nm) could be found with a quality factor of $\sim 930,000$.

Figure RL 5. Simulated resonance spectra of a 20 μm PEGDA microbead.

Comments #7: As the total upconversion emission is enhanced, how is the lifetime affected by the modification?

Response: As mentioned in the response to comment 2 (Figure RL 3), although the upconversion fluorescence of Eu^{3+} mediated UCNPs can achieve five orders of magnitude, negligible lifetime variation at 614 nm from Eu^{3+} could be observed when dielectric superlens was introduced. Thus, we can conclude that the significant enhancement of upconversion fluorescence induced by the dielectric superlensing effect is not due to the increase of the radiative rate of emitters.

Comments #8: In line 1, paragraph 2 of Page 2, the terminology ‘bidirectional wavefront modulation’ is not common in the field of optics and photonics.

Response: In the revised manuscript, we have changed ‘bidirectional wavefront modulation’ to ‘bidirectional light confinement’.

Comments #9: In Page 4, ‘We also observed that the enhancement effects of dielectric superlensing are strongly dependent on the specific nature of upconversion transitions.’ This is not clear to me.

Response: We thank this reviewer for pointing out this. Actually, as we mentioned in the response to comment 2, lanthanide-doped photon upconversion is quite a unique and nonlinear like optical process. For example, upconversion fluorescence governed by a 5-photon pumping process (Eu^{3+}) could be much more sensitive to the excitation power than that governed by a 4-photon pumping process (Tm^{3+}) (Figure 3a, b). Therefore, after introducing the dielectric superlenses, the enhancement for 5-photon governed upconversion fluorescence could be more evident than that governed by 4-photon pumping process.

To make it clear, in our revised manuscript, we have modified the description to 'We also observed that the upconversion fluorescence enhancement features of dielectric superlensing are strongly dependent on the upconversion population processes of different activators'.

Comments #10: In page 4, 'we quantified the enhancement factor for the characteristic emission peaks of each activator (Fig. 2d).' there maybe spectrum overlap/interference between different doping ions, how to separate them and eliminate the interference before quantification? How is Fig2d obtained?

Response: We thank this reviewer for raising this concern. Actually, due to the primary forbidden nature of the 4f–4f transition in lanthanide ions (*Chem. Rev.* **2014**, *114*, 5161), the emission bands of upconversion fluorescence are much narrower (FWHM < 20 nm) than those of conventional organic dyes (FWHM > 50 nm). In our work, although the emission bands of 450 nm and 540 nm have some overlap with their adjacent bands (Figure 2a), the interferences are quite small. For example, as shown in Figure RL 6, the peak fitting result showed that the interference from the adjacent emission band on the 450 nm emission peak is only less than 2%. Therefore, using the peak intensity of each emission band to quantify the upconversion fluorescence enhancement should be acceptable. We also hope the reviewer concurs.

Thus, Figure 2d was obtained by comparing the emission peak intensity of each emission band of the upconverting PDMS film with or without the dielectric microbead coverage. According to the reviewer's suggestion, details have been added in our revised manuscript.

On a separate note, the inset figure of Figure 2d has been removed to keep self-consistent and avoid repetition with data in Figure 3a.

Figure RL 6. Peak fitting for evaluating the interference from the adjacent emission band on the emission band (450 nm) of Tm^{3+} ions.

Comments #11: The method to quantify the enhancement factor for each activator is not clearly stated.

Response: As mentioned in the response to comment 10, we quantified the enhancement factor for each activator by dividing the peak intensity of its emission band by that of when microbeads were not used. Details has been added in the revised manuscript.

Comments #12: The authors should provide short explanation on simulation method in the main text rather than just state in the supplementary files. Why the method is used?

Response: Following the reviewer's suggestion, we have provided more details about the simulation in our revised manuscript.

Comments #13: Periodic boundary condition should be used, instead of 'FDTD simulation', to simulate the enhancement effect. Photonic crystal effect should be considered.

Response: Following the reviewer's suggestion, we have conducted the FDTD simulation using periodic boundary condition (Figure 1c). The simulation result is slightly changed when compared to that obtained using perfectly matched layer boundary condition. We also studied the photonic crystal effect of the dielectric superlensing array, as shown in Figure RL 4, no transmission bands were observed in the visible region. Therefore, the photonic crystal effect of the microbead array should play a minimum role in affecting the upconversion enhancement.

Comments #14: On page 6, it says 'Our control experiments involving poly (ethylene glycol) diacrylate microbeads with a mean diameter of 50 μm resulted in luminescence enhancement comparable to that of 20 μm -sized microbeads (Extended Data Fig. 7).' The authors should discuss about what is the point of the statement.

Response: We have revised this statement in the revised manuscript. Our further experiments involving poly(ethylene glycol) diacrylate microbeads with a mean diameter of $\sim 50 \mu\text{m}$ resulted in slightly inferior upconversion luminescence enhancement compared to that of 20 μm -sized microbeads (Supplementary Fig. 9). This might be due to incomplete polymerization for large-sized microbeads during UV-curing and the associated increase in focal length beyond the upconversion thin film.

Comments #15: The caption Fig 3c is not clear.

Response: We have revised the Fig 3c caption.

Comments #16: The reference should be added after the exact item rather than aggregate together after one sentence if several references are cited.

Response: Following the suggestion, we have made the changes.

In closing, we believe that the efficient dielectric superlensing strategy could help to take the UCNPs a big step forward in practical applications such as bioimaging, single-particle tracking, or even lasing. We also believe that by addressing reviewer's comments, we are now able to translate his/her critical input to a significant improvement of the manuscript, making it more reader-accessible and easy to understand by researchers from different disciplines.

Reviewer #3:

In this work, the authors propose an elegant method for enhancing the light intensity emitted by upconversion nanoparticles (UCNPs). Upconversion is a multiphoton process that has low probability and therefore that needs in principle high excitation power density. This aspect is hampering the use of the otherwise highly performant UCNPs in a wealth of applications including high-resolution bioimaging. Until now, several strategies have been tested to increase the luminosity of UCNPs, including plasmonics, broadband excitation, and optical crystals. In related fields, some researchers have developed local mini-lenses to concentrate the excitation beam (e.g. solar pumped YAG lasers, see Motohiro et al., Jap. J. Appl. Phys. 2017, DOI) or when it comes to microscopy, super lenses in order to overcome the diffraction limit and to reach resolution on the order of 100 nm (see Bing Yan et al., Appl. Optics 2017, 46, 3142). Here the authors transpose these ideas in the microworld by developing polymeric microbeads generating a superlensing effect. Irradiation of the beads with a NIR light results in a concentrated light beam (< 1 micrometer in diameter) that efficiently excites underlying UCNPs. In turn, the upconverted light can be used as a light source for exciting downshifted emission of other lanthanide ions, lending to the system a welcome versatility. The design is clever and the authors convincingly demonstrate that it indeed works and that the intensity enhancement obtained for the upconversion light reaches five orders of magnitude. This is one order of magnitude larger than previously reported devices. Overall the microbead/UCNPs proposed systems are adequately characterized and their properties investigated with respect to several parameters. Convincing examples are also given. As conclusion I think that this work is worth publication in Nature Communications after the following minor points are taken into consideration:

Response: We thank the reviewer for the time spent as well as positive comments on our work.

Comments #1: In the discussion section, it would be nice could the authors compare their proposed system with the enhancement systems published in the literature, not only with respect to performances but, also, with respect to ease of design and practical use.

Response: We thank the reviewer for this valuable comment. Following the suggestion, we have added more discussion in the discussion section of the revised manuscript to show the advances of our dielectric superlensing strategy mediated upconversion fluorescence enhancement.

Comments #2: Page 2, Last section, 2nd sentence. Please rephrase this sentence; firstly, green and orange light are not intrinsically “invisible”; what the authors mean is that they use a low power so that these beams are not detectable by naked eye. Secondly, I do not understand how these two beams “self-assemble”; this is not a correct way of describing the interaction between two light beams. Indeed self-assembly means that two or more components interact to give a more complex architecture. It is not the case here; self-assembly cannot qualify the interaction of two light beams – if they interact at all, because the generated bright spots (visible to the naked eye?) seem to have the same color than the initial beams.

Response: We thank the reviewer for careful reading of our manuscript. In the revised manuscript, the sentence has been rephrased to “Obviously, when two crossed LED light beams with green and orange color emission shined upon the microbeads obliquely, these beams passing through the microbeads converged to two small and bright spots on the substrate.” to make it clear and accurate.

Comments #3: Figure 2a, ED Figure 2 and corresponding text: I have some difficulty with qualifying the luminescence of europium as being “upconverted light”. The original definition of upconversion calls for a process in which an atom (ion) absorbs two (or more) low-energy photons and generates an anti-Stokes emission of one higher energy photon. This is obviously not the case for Tm@Eu in which the upconversion occurs for Tm(III) which then transfers the excitation energy to Gd(III) and Eu(III). To avoid some confusion, the authors should clearly state that they use an extension of the upconversion concept, in that the emitting ion is not the one absorbing two (or more) photons. Or more correctly that Eu(III) emission is activated by the up-converted light of Tm(III).

Response: We thank the reviewer for this suggestion. Actually, the emission from Eu^{3+} is a combination of upconversion and downshifting processes. Firstly, upconversion energy from Tm^{3+} could be transferred to Gd^{3+} ions, then the energy from Gd^{3+} could be downshifted to Eu^{3+} ions in the shell layer to generate fluorescence. More details could be found from our previously published paper (*Nat. Mater.* 2011, 10, 968).

Following the suggestion, in our revised manuscript, we have added related description as below:

‘It should be noted that the Eu^{3+} emission stems from the energy-migration-mediated upconversion by accepting the down-shifting energy from Gd^{3+} sublattice which is pre-populated by Tm^{3+} activators in the core.’

Comments #4: Figures 3a,b. Mechanism of energy transfer: According to Figure 3a, the slope of the Tm-upconverted emission versus excitation power is about 4 as is the slope of the Eu(III) emission; therefore why in Figure 3b the mechanistic path is going through the Tm(116) level that necessitates the absorption of 5 photons? Can one also consider a radiative excitation from the Tm(1D2) level? What happen if Gd(III) is replaced with non-luminescent Lu(III)? What is the explanation for the slope diminishing considerably (to around 1.5) when microbead coverage is introduced?

Response: Although lanthanide-doped upconversion fluorescence is nonlinear like and sensitive to the excitation power density, the slope could be quickly reduced when the excitation intensity is increased (*Phys. Rev. B.* 2000, 61,3337; *Phys. Rev. B.* 2005, 71, 125123). This is because that for a certain emitting state its excited state depopulation path could be gradually dominated by further population to higher excited states instead of relaxation to ground state for emitting. Besides, only under very low excitation power, the value of slope could be close to the number of photons involved in upconversion process. For example, for a 5-photon governed upconversion process, the slope could be very close to 5 upon very low power excitation, however, this value could be quickly reduced to 2 or even close to 1 when the excitation power is increased. More discussion has been added in the revised manuscript.

In our work, although the slope for 614 nm emission from Eu^{3+} is about 4.4, this value could further increase and finally approach to almost 5 if we keep reducing the excitation power for testing. We can also find this phenomenon from our simulation results (Figure 3c). For a 5-photon upconversion process, the slope is 5 at the initial stage, however this value would decrease when the pumping rate is increased.

As the introduction of microbeads could help to create photonic hotspots with very high local excitation power, although the output power of laser source is not changed, UCNPs under these microbeads would meet a much higher pumping photon flux, thus resulting in a diminished slope.

According to our previous investigation (Figure RL 7, *Nat. Mater.* 2011, 10, 968), if Gd^{3+} is replaced with nonluminescent Lu^{3+} or Y^{3+} , the fluorescence from $\text{Tb}^{3+}/\text{Eu}^{3+}$ could be significantly reduced. Although we also believe that the energy transfer from Tm^{3+} to $\text{Tb}^{3+}/\text{Eu}^{3+}$ can occur, no noticeable emission was observed when

Gd³⁺ in the core was replaced by Y³⁺. Thus, we can infer that the contribution from energy transfer from Tm³⁺ to Tb³⁺/Eu³⁺ is negligible.

Figure RL 7. Emission spectra of the core/shell nanoparticles showing the suppressed Tb³⁺ emission at reduced Gd³⁺ concentrations in the core level, demonstrating the key role of the Gd sublattice in transferring the energy to the activator. (*Nat. Mater.* 2011, 10, 968)

Comments #5: ED Figure 6a and associated text page 5. “Down-conversion” should not be used here in that it is defined as the “inverse” of upconversion (absorption of a high-energy photon leading to emission of two or more lower-energy photons). “Down-shifting” would be the appropriate term.

Response: We appreciate this comment. Following the suggestion, we have changed term ‘down-conversion’ to ‘down-shifting’ throughout the whole revised manuscript.

Comments #6: Surface power unit. Four different units are used throughout the manuscript which may be confusing, e.g. W/cm²; mW/cm²; W/mm²; mW/mm²; at least the latter two should be replaced with the first two. In particular, p. 7, while discussing optogenetic applications, an excitation power of 110 mW/mm² is mentioned; this translates into 11 W/cm², a value far above the accepted safe range for biological tissues.

Response: According to the reviewer’s suggestion, W cm⁻² was used as the unit throughout the revised manuscript.

We understand the reviewer’s concern about the excitation power above the safety range for biological tissues. Benefiting from the deep tissue penetration capability of NIR light, lanthanide-doped UCNPs have been extensively studied in optogenetics recently (*Science*, 2018, 359, 679; *Sci. Rep.*, 2015, 5, 16533; *ACS Nano*, 2016, 10, 1060). However, due to the low quantum yield of UCNPs, a strong laser source is usually used to stimulate UCNPs in cells or tissues. In the work of Chen *et al.* (*Science*, 2018, 359, 679), the power density used was 822 W cm⁻² to activate ChR2-VTA neurons in the brain slice. Furthermore, as compared to earlier upconversion NIR optogenetics publications, the power used was significantly higher at 4000 W cm⁻². (*Sci. Rep.*, 2015, 5, 16533). Therefore, benefiting from the dielectric superlensing strategy, our power used at 11 W cm⁻² is much lower than the current NIR-optogenetics standard. In addition, during testing, similar to two-photon imaging, pulse light instead of continuous excitation was used, thus the potential overheating effect could be further eliminated by lowering the frequency for excitation. In conclusion, we believe that the required excitation power could be further reduced by developing much brighter UCNPs in the near future.

Reviewers' comments:

Reviewer #1 (Remarks to the Author):

The quality of this revised manuscript, supported by additional experiments and discussions, have been significantly improved. I do not have additional questions.

Reviewer #2 (Remarks to the Author):

Although the manuscript has been revised, there are serious inconsistencies in the manuscript and the response letter.

In the response letter "Fig. RL 5", the authors present the simulated response spectra of the microbead with a quality factor of $\sim 930,000$. If depositing upconversion nanoparticles onto the microbead with such a high-quality factor, the corresponding lifetime will be definitely reduced due to the Purcell factor enhancement. However, this simulation result is contradicting with the measured emission lifetime of upconversion nanoparticles with and without microbeads as shown in Figure RL 3.

In Fig. 1(c), the authors present the simulation result on the enhanced intensity $|E|^2$ enhancement factor of 312 for the electrical intensity. This simulated enhancement factor is not consistent with the measured one-photon fluorescence enhancement factor of ~ 8 as shown in Fig. 3(e) and Fig. 3(f).

In addition, Fig. 3(f) and Fig. 3(g) are not consistent with each other. In Fig. 3(f), the enhancement factor is ~ 8 for the one-photon fluorescence; normally, the corresponding enhancement factor for the two-photon fluorescence is $\sim 8^2$ (i.e. 64). However, the enhancement factor for the two-photon fluorescence as shown in Fig. 3(g) is surprisingly >1000 .

The typical quantum yield of upconversion nanoparticle is 0.3%. In this manuscript, the authors claim that the enhancement factor is 103,000 for a 5-order upconversion. Will the quantum yield be higher than 100% for the upconversion nanoparticles with microbeads, given that the same light source is used for excitation?

Thus, given these serious inconsistencies in this manuscript, I do not recommend the editor to accept this manuscript.

Reviewer #3 (Remarks to the Author):

In this revised version, the authors have taken into account the numerous remarks of the three reviewers and modified it accordingly while adding some new experiments. On my side, I am happy with the changes and have no further comment. I recommend publication.

Review Matrix

<Responses to the reviewers' comments (Manuscript number NCOMMS-18-23530A)

Title: Upconversion Amplification through Dielectric Superlensing Modulation

Reviewer(s)' Comments to Author:

Referee: 3

Comments to the Author

Changes in the revised manuscript as a response to the reviewers' comments are highlighted in red color and clarifications regarding the reviewer's comments are provided in blue color.

Reviewer #1:

The quality of this revised manuscript, supported by additional experiments and discussions, have been significantly improved. I do not have additional questions.

Response: We are grateful for this reviewer's favorable comments.

Reviewer #2:

Although the manuscript has been revised, there are serious inconsistencies in the manuscript and the response letter.

Comments #1: In the response letter "Fig. RL 5", the authors present the simulated response spectra of the microbead with a quality factor of $\sim 930,000$. If depositing upconversion nanoparticles onto the microbead with such a high-quality factor, the corresponding lifetime will be definitely reduced due to the Purcell factor enhancement. However, this simulation result is contradicting with the measured emission lifetime of upconversion nanoparticles with and without microbeads as shown in Figure RL 3.

Response: We thank this reviewer for the critical comment. With due respect, we feel that the reviewer misunderstood our experiments. For the simulation of the quality factor of the microbead, dipole sources were placed in the microbead, and a quality factor for whispering gallery mode of $\sim 930,000$ was obtained. We agree with the reviewer that "If depositing upconversion nanoparticles onto the microbead with such a high-quality factor, the corresponding lifetime will be definitely reduced due to the Purcell factor enhancement." However, in our study for the measurement of upconversion luminescence, UCNPs were neither embedded in the microbeads nor deposited on the surface of the microbeads. As shown in Figure 1c, UCNPs were dispersed in a PDMS film beneath the microbeads, and the emitting UCNPs in the focal spot is more than $3 \mu\text{m}$ away from the surface of the microbead. As emphasized in many previously published results (*Nat. Photonics*, 2011, 5, 738; *Nat. Photonics*, 2012, 6, 459; *Nat. Photonics*, 2014, 8, 835; *Sci. Rep.* 2014, 4, 6396), to ensure efficient Purcell effect, the emitters should be placed very close to ($<20 \text{ nm}$) the nanoantenna or attached on the surface of the cavity. Therefore, due

to the large distance between emitting UCNPs and the microbead ($>3 \mu\text{m}$), we believe the Purcell effect, if existing, in the microbead should be extremely weak on the spontaneous emission processes of UCNPs in the PDMS film. We have confirmed that the emission decay lifetime of UCNPs was not noticeably affected. To clarify this, we have included a discussion in the revised manuscript.

Comments #2: In Fig. 1(c), the authors present the simulation result on the enhanced intensity $|E|^2$ enhancement factor of 312 for the electrical intensity. This simulated enhancement factor is not consistent with the measured one-photon fluorescence enhancement factor of ~ 8 as shown in Fig. 3(e) and Fig. 3(f).

Response: We understand the reviewer's concern. It should be noted that when a microbead was placed on the top of a PDMS film embedded with one-photon emitters (downshifting nanoparticles or dye), only a small region beneath the microbead could be illuminated. Thus the emitters located in the area shadowed by the microbead will not contribute to the total luminescence intensity. For an example, if the area of focal spot is exactly 1% of the projected area of the microbead and the average excitation power density is enhanced by a factor of 100, then the total emission intensity for one-photon emitters would not be overall enhanced if the emission light collection effect from the microbead is not taken into consideration. This is because although luminescence from these 1% emitters is 100 times enhanced, the rest 99% emitters are not excited. However, according to our simulation, the dielectric microbead could contribute ~ 8 times enhancement in light collection efficiency, thus for downshifting nanoparticles or rhodamine dye, we obtained a very close one-photon fluorescence enhancement factor of ~ 8 . We have added a discussion in the revised manuscript to clarify this.

Comments #3: In addition, Fig. 3(f) and Fig. 3(g) are not consistent with each other. In Fig. 3(f), the enhancement factor is ~ 8 for the one-photon fluorescence; normally, the corresponding enhancement factor for the two-photon fluorescence is $\sim 8^2$ (i.e. 64). However, the enhancement factor for the two-photon fluorescence as shown in Fig. 3(g) is surprisingly >1000 .

Response: This concern is similar to the above one. For a two-photon nonlinear process, $I \propto P^2$, where P is the excitation power. We agree with the reviewer that if one-photon fluorescence is enhanced by 8 times, its two-photon absorption fluorescence will be enhanced by ~ 64 . However, as mentioned in the last example, the 8 times enhancement in one-photon fluorescence is from the light collection effect of microbeads, and the power density is enhanced by 100 times instead of 8. For the same example, if the area of the focal spot is 1% of the projected area of the microbead and the average excitation power density P in the focal spot was 100 times enhanced accordingly. In this case, the emission intensity for those 1% two-photon emitters in the focal spot would be enhanced to 10,000 (100^2) times. Although the rest 99% emitters are not in action, the overall emission would be 100 times stronger and a value about 800 (100×8) could be expected when the light collection effect was taken into consideration.

Comments #4: The typical quantum yield of upconversion nanoparticle is 0.3%. In this manuscript, the authors claim that the enhancement factor is 103,000 for a 5-order upconversion. Will the quantum yield be higher than 100% for the upconversion nanoparticles with microbeads, given that the same light source is used for excitation?

Response: Usually, Er^{3+} activated bulk or core-shell UCNPs (2-3 photon processes) show much higher quantum yield of $\sim 0.3\%$ upon strong excitation (150 W cm^{-2}) and widely used in many emerging applications (*Nanoscale*, 2010, 2, 1417). However, it should be noted that the quantum yield of UCNPs is highly sensitive to the excitation power due to the nonlinear power-dependence of upconversion luminescence, and a slight decrease in excitation power could lead to significant decrease in upconversion fluorescence and thus the quantum yield. For example,

for a 3-photon or 5-photon mediated upconversion process, the upconversion fluorescence would be reduced to $1/8$ (0.5^3) or $1/32$ (0.5^5) of the original intensity when the excitation power is reduced into half. Thus, for upconversion processes involving more pumping photons especially energy migration mediated upconversion which is governed by 5-photon process, the quantum yield could be extremely low under low power excitation. In addition, without the inert shell protection, the upconversion luminescence would suffer severe surface quenching, leading to further reduced quantum yield. For example, as reported in *Nat. Photonics*, 2018,12, 154, although the 3-photon upconversion emission intensity of core UCNPs ($\text{NaYF}_4:\text{Yb/Tm}$) was enhanced about 2,000 folds, the overall quantum yield is still only $\sim 3.6\%$ (10 W cm^{-2}), thus we can infer a quantum yield of $\sim 0.0018\%$ of the control sample. Therefore, the quantum yield of UCNPs could vary in an extremely wide range and not just around 0.3% .

In our experiment, under a much lower excitation power of 1.5 W cm^{-2} , $\text{NaGdF}_4:\text{Yb/Tm}@\text{NaGdF}_4:\text{Eu}$ is governed by a 5-photon process and the Eu^{3+} activators doped in the shell layer are not protected by an inert layer, all these factors would contribute to a much lower quantum yield, thus providing excellent room for further enhancement. In this case, as mentioned in response to comment #2 and #3, the introduction of dielectric microbeads could help to create local excitation regions with much higher excitation power density, and upconversion luminescence of UCNPs in focal spots could be significantly enhanced, resulting in an enormous enhancement in overall luminescence intensity.

Reviewer #3:

In this revised version, the authors have taken into account the numerous remarks of the three reviewers and modified it accordingly while adding some new experiments. On my side, I am happy with the changes and have no further comment. I recommend publication.

Response: We thank this reviewer's positive comments.

Reviewers' comments:

Reviewer #2 (Remarks to the Author):

1. For the completeness of the information, the authors should include "Fig. RL 5" and the corresponding descriptions into the supporting information of the manuscript, for the convenience of the future readers.

The reference papers provided by the authors are not related to the coupling of quantum emitters close to Whisper Galley Mode. For example, "Nat. Photonics, 2011, 5, 738; Nat. Photonics, 2012, 6, 459; Nat. Photonics, 2014, 8, 835" are related to the emitters as coupled to plasmonic nanocavity. "Sci. Rep. 2014, 4, 6396" is on the coupling of Whisper galley mode from free space into the cavity using the Rayleigh scattering of the nanoparticle, but not the emission characteristic of the quantum emitters close to Whisper Gallery mode.

In order to have a strong evidence to clarify the role of the Purcell factor enhancement, the authors should add the FDTD simulation results of the Purcell factor when a dipole source is placed at different locations with respect to the surface of the microbeads (based on the schematic in Fig. 1c). The method to simulate Purcell factor for dipole source is well established (see details in https://kb.lumerical.com/en/diffractive_optics_cavity_purcell_factor.html). The authors should include this simulation results in the supporting information as well.

2. Regarding the previous Comment 2 and Comment 3, the author claims that the enhanced emissions in Fig. 3(e) and Fig. 3(f) are due to the ~8 times enhancement in light collection efficiency. Based on the information as given by the authors in the response letter, the area of the focal spot is 1% of the projected area, and the rest 99% emitters are not in action.

I strongly disagree with the authors' explanation.

The enhancement factor for the pumping power during 1 photon process could be denoted as "N1p", and the enhancement factor for the 2-photon process could be denoted as "N1p × N1p".

By following the argument as given by the authors, during the 1-photon process, the resultant enhancement during the pumping process is given by:

$$\text{Enhancement Factor (one-Photon)} = 1\% \times N1p + 99\% \times 0 \quad (1)$$

During the 2-photon process, the resultant enhancement during the pumping process is given by:

$$\text{Enhancement Factor (Two-Photon)} = 1\% \times N1p \times N1p + 99\% \times 0 \times 0 \quad (2)$$

Based on the measurement results as shown in Fig. 3(e) and Fig. 3(f), at the lowest pump power of 1.5 W cm⁻², the measured enhancement factors for one-photon process and two-photon process are around ~7.8 and ~1400. In other words, "Enhancement Factor(one-Photon)" is ~7.8 and "Enhancement FactorTwo Photon" is ~1400.

Then, by Dividing Eq. (2) over Eq. (1) and then substituting these two numbers, we can obtain the value for "N1p", which is ~179. Now, based on this number "N1p = ~179", the enhancement factor during the pumping process can be calculated by

$$\text{Enhancement Factor(one-Photon)} = 1\% \times N1p = 179\% \quad (3).$$

In other words, the upconversion nanoparticle is experiencing an optical power, which is higher than the total pump power. How can it be possible?

Due to the inconsistencies in the manuscript, I don't think this manuscript suitable to publish in Nat Comm.

Reviewer #3 (Remarks to the Author):

To me, most of the reviewers' comments have been answered satisfyingly. Here I am only considering the matter of enhancement raised by reviewer 2.

Report 1

a) Comment #3. It seems that the reviewer has indeed overseen the power density enhancement, so that the 2-photon enhancement becomes much larger than 64 and the authors' estimate of a potential enhancement factor of 800 is reasonable. However, the authors have to explain why the measured enhancement factor is >1000, especially that at higher power density excitation, the square relationship has the tendency to level off and even to slowly go back to a linear dependence at much larger power densities.

b) Comment #4. It occurs to me that there is confusion between quantum yield and emission intensity. The quantum yield increases with increasing power density but light collection also increases, therefore the brightness increases more than the quantum yield. The authors should definitively clarify these points, carefully writing which quantity is affected by the "enhancement". Latest report

c) I do not quite understand the calculations made by the reviewer. Experimentally, N_{1p} is determined to be around 8 and N_{2p} around 1400. There is no point in dividing the two equations (2)/(1) to retrieve N_{1p} since N_{2p} includes the enhanced intensity arising from both the 8-fold improved absorption and from the increase in power density excitation! There is no difference with Comment #3 above.

In conclusion, I think that the authors should develop this aspect much more carefully, and clearly state the various contributions to the overall brightness of their UCNPs (data reported in Figure 3) so that their claim about the "enhancement factor" becomes trustable.

Review Matrix

<Responses to the reviewers' comments (Manuscript number NCOMMS-18-23530B)

Title: Upconversion Amplification through Dielectric Superlensing Modulation

Reviewer(s)' Comments to Author:

Referee: 2

Comments to the Author

Changes in the revised manuscript as a response to the reviewers' comments are highlighted in red color and clarifications regarding the reviewer's comments are provided in blue color.

Reviewer #2:

Comments #1: For the completeness of the information, the authors should include “Fig. RL 5” and the corresponding descriptions into the supporting information of the manuscript, for the convenience of the future readers.

The reference papers provided by the authors are not related to the coupling of quantum emitters close to Whisper Galley Mode. For example, “Nat. Photonics, 2011, 5, 738; Nat. Photonics, 2012, 6, 459; Nat. Photonics, 2014, 8, 835” are related to the emitters as coupled to plasmonic nanocavity. “Sci. Rep. 2014, 4, 6396” is on the coupling of Whisper galley mode from free space into the cavity using the Rayleigh scattering of the nanoparticle, but not the emission characteristic of the quantum emitters close to Whisper Gallery mode.

In order to have a strong evidence to clarify the role of the Purcell factor enhancement, the authors should add the FDTD simulation results of the Purcell factor when a dipole source is placed at different locations with respect to the surface of the microbeads (based on the schematic in Fig. 1c). The method to simulate Purcell factor for dipole source is well established (see details in https://kb.lumerical.com/en/diffractive_optics_cavity_purcell_factor.html). The authors should include this simulation results in the supporting information as well.

Response: We thank the reviewer's kind suggestion. Fig. RL 5 has been updated in Supplementary Fig. S6a. Following the reviewer's suggestion, we have carried out simulations to study the Purcell factors for a dipole source placed at different locations with respect to the microbead surface in the revised manuscript. As shown in Fig. RL 1b (also included in Supplementary Fig. S6), Purcell factors are very close to one when the dipole is placed at 0.1 to 5 μm away from the microbead surface. The negligible influence from the Purcell effect on radiative emission rate enhancement was further confirmed by the constant luminescence decay lifetime across the whole Eu^{3+} emission band from 610 to 620 nm (1 nm step for each measurement) (Fig. RL 1c, d). More importantly, no lasing-like sharp emission peaks were detected in the enhanced upconversion emission spectra (Fig. RL 1d inset). Therefore, we can infer that due to the large distance between the emitting UCNPs in the focal spot and the microbead in our current system, the Purcell effect shows negligible contribution to the overall upconversion fluorescence enhancement. Related description has also been added in the manuscript and Supplementary Information.

Figure RL 1. (a) Simulated resonance spectra of a microbead with a dipole source attached on its surface. (b) Purcell factors of a dipole source placed at different locations (0.1 to 5 μm) with respect to the surface of the microbead. (c) Upconversion luminescence lifetimes of Eu^{3+} at 614 nm (emission peak) with and without the microbead. (d) Upconversion luminescence lifetimes of Eu^{3+} across the whole emission band from 610 to 620 nm with 1 nm-step for each measurement. Inset: emission spectra of Eu^{3+} mediated photon upconversion with and without the microbead.

Comments #2: Regarding the previous Comment 2 and Comment 3, the author claims that the enhanced emissions in Fig. 3(e) and Fig. 3(f) are due to the ~ 8 times enhancement in light collection efficiency. Based on the information as given by the authors in the response letter, the area of the focal spot is 1% of the projected area, and the rest 99% emitters are not in action. I strongly disagree with the authors' explanation. The enhancement factor for the pumping power during 1 photon process could be denoted as "N1p", and the enhancement factor for the 2-photon process could be denoted as "N1p \times N1p". By following the argument as given by the authors, during the 1-photon process, the resultant enhancement during the pumping process is given by: Enhancement Factor (one-Photon) = 1% \times N1p + 99% \times 0 (1) During the 2-photon process, the resultant enhancement during the pumping process is given by: Enhancement Factor (Two-Photon) = 1% \times N1p \times N1p + 99% \times 0 \times 0 (2) Based on the measurement results as shown in Fig. 3(e) and Fig. 3(f), at the lowest pump power of 1.5 W cm^{-2} , the measured enhancement factors for one-photon process and two-photon process are around ~ 7.8 and ~ 1400 . In other words, "Enhancement Factor(one-Photon)" is ~ 7.8 and "Enhancement Factor Two Photon" is ~ 1400 . Then, by Dividing Eq. (2) over Eq. (1) and then substituting these two numbers, we can obtain the value for "N1p",

which is ~179. Now, based on this number “ $N_{1P} \approx 179$ ”, the enhancement factor during the pumping process can be calculated by Enhancement Factor(one-Photon) = $1\% \times N_{1P} = 179\%$ (3). In other words, the upconversion nanoparticle is experiencing an optical power, which is higher than the total pump power. How can it be possible?

Response: We appreciate this reviewer’s effort. However, I believe this review has miscalculated the enhancement factor.

Enhancement Factor (one-Photon) = $1\% \times N_{1P} + 99\% \times 0$ (1)

Enhancement Factor (Two-Photon) = $1\% \times N_{1P} \times N_{1P} + 99\% \times 0 \times 0$ (2)

According to the well-known formula $I \propto P^n$, we think the N_{1P} in above formulas should be replaced by the enhancement factor (**E**) of the local excitation power density (**P**) instead of the overall enhancement factor (N_{1P}) of the upconversion fluorescence. Here, for simplicity purpose, we assume that the area of focal spot is exactly 1% of the projected area of the microbead and the local excitation power density is thus enhanced by a factor of 100 (**E** = 100). Besides, microbeads could also contribute additional enhancement (~8) to the overall upconversion emission through efficient light collection.

So, the overall upconversion fluorescence enhancement factor could be estimated as below:

Enhancement Factor (One-Photon) $N_{1P} = (1\% \times E + 99\% \times 0) \times 8 = 8$ (1)

Enhancement Factor (Two-Photon) $N_{2P} = (1\% \times E^2 + 99\% \times 0 \times 0) \times 8 = 800$ (2)

Now, if we divide (2) over (1), the value can be obtained is the enhancement factor (**E**) for local excitation power density instead of overall one-photon fluorescence enhancement (N_{1P}).

Here the reason we assume the 100-fold enhancement of excitation power density after passing through the microbead is just for simplicity purpose. In real case, the microbead could provide a higher enhancement in excitation power density, meaning that **E** with a higher value than 100 should be used in above formulas. For example, by dividing 1400 (N_{2P}) over 7.8 (N_{1P}), an **E** value of ~ 179 could be obtained.

Thus, although the formulas used here are simplified, the estimated results could still reflect the consistency of our experimental data. We hope the referee concur.

Reviewer #3:

To me, most of the reviewers’ comments have been answered satisfyingly. Here I am only considering the matter of enhancement raised by reviewer 2.

- a) Comment #3. It seems that the reviewer has indeed overseen the power density enhancement, so that the 2-photon enhancement becomes much larger than 64 and the authors’ estimate of a potential enhancement factor of 800 is reasonable. However, the authors have to explain why the measured enhancement factor is >1000, especially that at higher power density excitation, the square relationship has the tendency to level off and even to slowly go back to a linear dependence at much larger power densities.

Response: We would like to thank this reviewer for the support. As we clarified in the response to comment #2 from referee 2, the reason we assume the 100-fold enhancement of excitation power density after passing through the microbead is just for simplicity purpose. In real case, the microbead could provide a higher enhancement in excitation power density, thus resulting in an overall enhancement factor higher than 800.

Lanthanide-doped upconversion nanomaterial is a unique type of nonlinear-optical materials. As illustrated in Figure RL 2, by accepting energy from sensitizers those harvesting NIR photons, lanthanide activators could be sequentially populated to higher excited states through intermediate excited states. In principle, the intensity of emission I arising from a n -photon populated excited state increases as a power function of excitation power P ($I \propto P^n$) at the low-power excitation. However, when the excitation power is increased, instead of returning back to the ground state for emitting, activators in the current state ($n = 2$ for example) are highly probable to be further populated to much higher emitting states (n_3), thus the upconversion emission intensity from this n -photon state (n_2) will gradually tend to reach saturation along with the excitation power increase (see *Phys. Rev. B* 61, 3337-3346 (2000)). Related clarification has been highlighted in our manuscript. We hope our clarification is clear.

Figure RL 2. Working principle of a typical energy-transfer upconversion process.

- b) Comment #4. It occurs to me that there is confusion between quantum yield and emission intensity. The quantum yield increases with increasing power density but light collection also increases, therefore the brightness increases more than the quantum yield. The authors should definitely clarify these points, carefully writing which quantity is affected by the “enhancement”.

Response: We understand the reviewer’s concern. The quantum yield is the conversion efficiency of photons absorbed to photons emitted by the sample. Since upconversion emission intensity is much easier to be obtained during fluorescence measurement, we thus use the variation of emission intensity (peak counts) to evaluate the enhancement factor of upconversion luminescence throughout the whole manuscript. Related contents are highlighted in our manuscript. In our system, the overall upconversion emission enhancement is contributed by two factors. One is the significantly enhanced local excitation power density, and the other one is the enhanced far-field collection efficiency of emitted photons. We agree with the referee that the emission intensity increases more than the quantum yield due to the light collection effect of microbeads. Therefore, although both the quantum yield and emission intensity could be affected by the “enhancement”, we think the investigation of enhancement in “emission intensity” is more valuable for practical application since the overall upconversion output is what we usually care. We hope the reviewer concur.

- c) I do not quite understand the calculations made by the reviewer. Experimentally, N_{1p} is determined to be around 8 and N_{2p} around 1400. There is no point in dividing the two equations (2)/(1) to retrieve N_{1p} since N_{2p} includes the enhanced intensity arising from both the 8-fold improved absorption and from the increase in power density excitation! There is no difference with Comment #3 above.

Response: We thank the reviewer's further explanation. We have pointed out the same issue in the response to reviewer #2. We have also modified the formulas and evidenced the consistency of our experimental data as below.

According to the well-known formula $I \propto P^n$, we think the N_{1p} in above formulas should be replaced by the enhancement factor (**E**) of the local excitation power density (**P**) instead of the overall enhancement factor (N_{1p}) of the upconversion fluorescence. Here, for simplicity purpose, we assume that the area of focal spot is exactly 1% of the projected area of the microbead and the local excitation power density is thus enhanced by a factor of 100 (**E** = 100). Besides, microbeads could also contribute additional enhancement (~ 8) to the overall upconversion emission through efficient light collection.

So, the overall upconversion fluorescence enhancement factor could be estimated as below:

Enhancement Factor (**One-Photon**) $N_{1P} = (1\% \times E + 99\% \times 0) \times 8 = 8$ (1)

Enhancement Factor (**Two-Photon**) $N_{2P} = (1\% \times E^2 + 99\% \times 0 \times 0) \times 8 = 800$ (2)

Now, if we divide (2) over (1), the value can be obtained is the enhancement factor (**E**) for local excitation power density instead of overall one-photon fluorescence enhancement (N_{1p}).

In conclusion, I think that the authors should develop this aspect much more carefully, and clearly state the various contributions to the overall brightness of their UCNPs (data reported in Figure 3) so that their claim about the "enhancement factor" becomes trustable.

Response: We thank the reviewer's effort on our manuscript. We have clarified the various contributions to the overall enhancement of UCNPs and related issues in our manuscript and hope the current version is much clearer.

REVIEWERS' COMMENTS:

Reviewer #2 (Remarks to the Author):

Comments #(a):

In the latest response letter (figure RL 1b.), the authors only plot out 5 points with the selected distance values (i.e. 100 nm, 500 nm, 1 μm , 3 μm , and 5 μm). In fact, it is not appropriate to link these 5 points with a straight line, because the distance interval between these selected points is even as large as 2 μm , which is more than 3 times of the emission wavelength at 610-620 nm.

I suggest the authors to add the following simulations in figure RL 1b. in order to have a comprehensive conclusion:

1. To plot out the Purcell factors with the distance between the dipole source and the microbead surface, from 10 nm to 5 μm , with a step of 10 nm.
2. To add simulation results for both the x-polarized dipole and the z-polarized dipole sources.
3. Experimentally characterize the effective dipole polarization for the upconversion nanoparticles placed near the microbead sphere at the emission wavelength 610-620 nm.

Now, we follow the authors' calculations, by dividing Eq. (4) over Eq. (3). The enhancement factor (E) for the local excitation power density is ~ 179 . In other words, the microbead sphere is able to focus the incident optical field onto microbead sphere, down to an area which is 1% of the projected area of the microbead, with the 179-time enhanced power density. Then, the total power could be calculated by using the relationship:

Total power = power density * area. (5)

Based on Eq. (5), the optical power after the microbead sphere could be calculated by multiplying the 1% area and the 179 time enhanced optical power density. Thus, the optical power after the microbead sphere is then 179%, as compared the incident optical power. It means that the microbead sphere itself can generate an additional 79% optical power. How can it possible? It violates the physical law of energy conservation.

Comments #(c):

As suggested by the Editor, I read through the response to Reviewer 3. Reviewer 3 indeed has the same concern whether the high "enhancement factor" claim is trustable or not.

To be more specific, in Question #3(a) as raised by Reviewer 3, Reviewer 3 is also asking the same question on the clear physical origin of the measured enhancement factor of >1000 . However, the authors respond as below:

"the reason we assume the 100-fold enhancement of excitation power density after passing through the microbead is just for simplicity purpose. In real case, the microbead could provide a higher enhancement in excitation power density, thus resulting in an overall enhancement factor higher than 800."

Here, the authors were using the terms "for simplicity purpose". It needs strong and solid experimental evidences to support the claim rather than assumption. Experiments should be carried out seriously and carefully so as to make the claim convincing.

To maintain the prestigious quality and high standard of Nature Communications, I strongly recommend the rejection of this manuscript.

Reviewer #3 (Remarks to the Author):

In this second revision, that authors have adequately addressed the concerns I raised and modified the manuscript accordingly.

Overall the ms describes a powerful and easy-to-implement way of amplifying upconversion luminescence through inducing both superior light collection and creating hotspots with high excitation field intensity, henceforth boosting the upconversion efficiency. The dielectric microbeads developed here create a superlensing effect that can also be used for amplifying one- and two-photon Stokes emission. They are easy to prepare and their effectiveness is convincingly demonstrated on practical applications.

Review Matrix

<Responses to the reviewers' comments (Manuscript number NCOMMS-18-23530C)

Title: Upconversion Amplification through Dielectric Superlensing Modulation

Reviewer(s)' Comments to Author:

Referee: 2

Comments to the Author

Reviewer #2:

Comments #1: In the latest response letter (figure RL 1b.), the authors only plot out 5 points with the selected distance values (i.e. 100 nm, 500 nm, 1 μ m, 3 μ m, and 5 μ m). In fact, it is not appropriate to link these 5 points with a straight line, because the distance interval between these selected points is even as large as 2 μ m, which is more than 3 times of the emission wavelength at 610-620 nm.

I suggest the authors to add the following simulations in figure RL 1b. in order to have a comprehensive conclusion:

1. To plot out the Purcell factors with the distance between the dipole source and the microbead surface, from 10 nm to 5 μ m, with a step of 10 nm.
2. To add simulation results for both the x-polarized dipole and the z-polarized dipole sources.
3. Experimentally characterize the effective dipole polarization for the upconversion nanoparticles placed near the microbead sphere at the emission wavelength 610-620 nm.

Response: We thank the reviewer's suggestion. As shown in Fig. RL1a, more simulated data points were added. We understand that the reviewer hopes to see simulation results with very small step (10 nm), however this will take several months since each simulation takes several hours to complete. To address the reviewer's concern, we further reduced the simulation steps with values smaller than half of the emission wavelength, and the results for z-polarized dipole source were added in the new simulation. Our simulation results suggest that the Purcell effect does not play a role in the luminescence enhancement in our study regardless of the polarization and distance of dipole sources.

Besides, following the reviewer's suggestion, we studied the polarization properties of upconversion luminescence (610-620 nm) by placing an analyzer in front of the spectrometer. As shown in Fig. RL1b, upconversion luminescence from UCNPs beneath microbeads hardly show any polarization property. This is because of the fact that the as-synthesized UCNPs are uniformly doped and have a perfectly spherical shape. In addition, their surrounding is isotropic. Similar results have also been reported in other work (*J. Phys. Chem. C*, 2018, 122, 15666).

Therefore, by combing the experimental and simulation results, we can infer that the Purcell effect shows negligible contribution to the overall upconversion fluorescence enhancement reported in our paper.

Fig. RL1. **a**, Simulated Purcell factors of dipole sources with varied distances to the microbead. **b**, Investigation of polarization property of upconversion luminescence from Eu activated upconversion nanocrystals beneath the microbead.

Comments #2: Now, we follow the authors' calculations, by dividing Eq. (4) over Eq. (3). The enhancement factor (E) for the local excitation power density is ~ 179 . In other words, the microbead sphere is able to focus the incident optical field onto microbead sphere, down to an area which is 1% of the projected area of the microbead, with the 179-time enhanced power density. Then, the total power could be calculated by using the relationship:

$$\text{Total power} = \text{power density} \times \text{area}. \quad (5)$$

Based on Eq. (5), the optical power after the microbead sphere could be calculated by multiplying the 1% area and the 179 time enhanced optical power density. Thus, the optical power after the microbead sphere is then 179%, as compared the incident optical power. It means that the microbead sphere itself can generate an additional 79% optical power. How can it possible? It violates the physical law of energy conservation.

Response: The reviewer has completely misunderstood our calculation. The result of 179 is estimated from the experimental data based on the proposed formula and 100 (and 1%) is just our assumption to make the reviewer better understand our calculation. We should not directly perform calculation by linking two numbers.

We should note that the electric field distribution is not homogenous, and the intensity is different from point to point in three dimensions. Besides, the absolute relationship between the power intensity and upconversion emission intensity is still not established since the excitation source has its own power distribution. Thus, instead of using a simple formula to predict the absolute luminescence intensity, researchers can only use a so-called "average power density" to roughly estimate the trend of upconversion luminescence. Although this is a rough estimation, this method is widely accepted and frequently used to explain the working mechanism of lanthanide-mediated photon upconversion.

Comments #3: As suggested by the Editor, I read through the response to Reviewer 3. Reviewer 3 indeed has the same concern whether the high "enhancement factor" claim is trustable or not.

To be more specific, in Question #3(a) as raised by Reviewer 3, Reviewer 3 is also asking the same question on the clear physical origin of the measured enhancement factor of >1000 . However, the authors respond as below:

"the reason we assume the 100-fold enhancement of excitation power density after passing through the

microbead is just for simplicity purpose. In real case, the microbead could provide a higher enhancement in excitation power density, thus resulting in an overall enhancement factor higher than 800.”

Here, the authors were using the terms “for simplicity purpose”. It needs strong and solid experimental evidences to support the claim rather than assumption. Experiments should be carried out seriously and carefully so as to make the claim convincing.

Response: As we mentioned above, we obtained the value of 800 when assuming the average power density is enhanced by 100 folds. According to the simulation result (see Fig. 1c), the power density could be enhanced much more than 100 folds at the foci, thus it is reasonable to lead to a higher enhancement factor although the formula is not absolutely accurate.

Reviewer #3:

In this second revision, that authors have adequately addressed the concerns I raised and modified the manuscript accordingly.

Overall the ms describes a powerful and easy-to-implement way of amplifying upconversion luminescence through inducing both superior light collection and creating hotspots with high excitation field intensity, henceforth boosting the upconversion efficiency. The dielectric microbeads developed here create a superlensing effect that can also be used for amplifying one- and two-photon Stokes emission. There are easy to prepare and their effectiveness is convincingly demonstrated on practical applications.

Response: We are grateful for this reviewer’s favorable comments.